# Decarbonising the iron and steel sector for a 2 °C target using inherent waste streams

Yongqi Sun[1,2], Sicong Tian[2], Philippe Ciais [3], Zhenzhong Zeng [1✉], Jing Meng [4✉] & Zuotai Zhang [1,5✉]

The decarbonisation of the iron and steel industry, contributing approximately 8% of current global anthropogenic $CO_2$ emissions, is challenged by the persistently growing global steel demand and limitations of techno-economically feasible options for low-carbon steelmaking. Here we explore the inherent potential of recovering energy and re-using materials from waste streams, high-temperature slag, and re-investing the revenues for carbon capture and storage. In a pathway based on energy recovery and resource recycling of glassy blast furnace slag and crystalline steel slag, we show that a reduction of $28.5 \pm 5.7\%$ $CO_2$ emissions to the sectoral 2 °C target requirements in the iron and steel industry could be realized in 2050 under strong decarbonization policy consistent with low warming targets. The technological schemes applied to engineer this high-potential pathway could generate a revenue of US$35 ± 16 and US$40 ± 18 billion globally in 2035 and 2050, respectively. If this revenue is used for carbon capture and storage implementation, equivalent $CO_2$ emission to the 2 °C sectoral target requirements is expected to be reduced before 2050, without any external investments.

[1] School of Environmental Science and Engineering, Southern University of Science and Technology, 518055 Shenzhen, China. [2] School of Chemical Engineering, The University of Queensland, Brisbane, St Lucia QLD 4072, Australia. [3] Laboratoire des Sciences du Climat et de l'Environnement, UMR 1572 CEA-CNRS UVSQ, 91191 Gif sur Yvette, France. [4] The Bartlett School of Sustainable Construction, University College London, London WC1E 7HB, UK. [5] The Key Laboratory of Municipal Solid Waste Recycling Technology and Management of Shenzhen City, 518055 Shenzhen, China.
✉email: zengzz@sustech.edu.cn; jing.j.meng@ucl.ac.uk; zhangzt@sustech.edu.cn

**E**ffective decarbonisation of the industrial sector is a key wedge for meeting low climate warming targets[1,2]. The iron and steel industry is a particularly energy- and emission-intensive sector that accounts for ~8% of annual global anthropogenic $CO_2$ emissions, >2800 Mt $CO_2$ per year[3–5]. Two recent emissions scenarios to limit warming below 2 °C[4,5] proposed that the $CO_2$ budget of iron and steel emissions should be capped at 50 Gt between now and 2050. Meeting this budget is challenged by the persistently growing steel demand, which is expected to rise from 1.82 Gt-steel in 2020 to 2.55 Gt-steel in 2050[5,6], driven by urbanisation and industrialisation in non-OECD countries[7,8]. If the current iron and steel production proceeds without the implementation of $CO_2$ emission reduction or carbon capture and storage (CCS), the total emission budget in this sector by 2050 will exceed by a factor of two the limit proposed by refs. [4,5].

In recent decades, the iron and steel industry has reduced its energy intensity by 60%, and most recent iron and steel production sites operate close to a thermodynamic limit of ~20 GJ per tonne of crude steel[4,5,7], still leaving an emission gap with respect to the requirement for a sectoral 2 °C target in refs. [4,5]. Some production technologies began to be implemented to reduce $CO_2$ emission, like top gas pressure recovery turbines (TRTs) and combined cycle power plants (CCPPs)[9]. CCS remains a critical option that could ultimately achieve a deep reduction in $CO_2$ emissions from this sector[5,10], although its costs and externalities remain to be assessed in practical industrial cases for coupling CCS facilities with steel production. However, the high incremental cost of deploying CCS is still far from an acceptable level. Recently, Tian et al.[5] demonstrated that industrial decarbonisation is not necessarily as expensive as usually considered if CCS measures can be deeply integrated into manufacturing processes. Furthermore, the net-zero emission steel target in the long term like the EU Green Deal[11] necessitates more breakthrough technologies such as hydrogen- and green electricity-based metallurgy and smart carbon usage (Process integration and Carbon Valorisation, Carbon Capture and Usage-CCU, etc)[11–13].

In the current manufacturing process of iron and steel, producing one tonne of crude steel generates waste streams in a range of 250–300 kilograms of blast furnace slag (BFS) at temperatures of 1500–1600 °C and 100–150 kilograms of steel slag (SS) at temperatures of 1550–1650 °C[14–17]. The high-temperature slag contains high-degree exergy at the levels of thermal energy and material resources, offering a large internal recovery potential and the revenues that could be re-invested into CCS to lower its cost[18–20]. From the energy point of view, the energy carried by slag represents 10–15% of the total energy input in the iron and steel industry today[18,19]. If this heat could be recovered, the embedded energy could be turned into reductions in fossil fuel use and $CO_2$ emissions. From the resource point of view, both BFS and SS contain >40 wt.% CaO fluxed from limestone calcination[21,22], the recycling of which constitutes a significant Ca-source, e.g., for slagmaking in metallurgy[23–28] and for cement production to reduce their $CO_2$ emissions[29–31].

In this work, we quantify the potentials for reusing heat and recycling iron and steel wastes rich in CaO as a feedstock for the cement industry based on technically and economically feasible solutions. We construct nine pathways in the timeframe of 2020–2050 based on fundamental properties of BFS/SS and technological levels. For the pathways with the highest-potential $CO_2$ emission reduction, we propose five engineering schemes and conduct a techno-economic analysis. We find that net revenues of US\$35 ± 16 billion and US\$40 ± 18 billion could be generated globally in 2035 and 2050. A re-investment of these revenues into CCS coupled to manufacturing processes could reduce equivalent $CO_2$ emissions before 2050 to be consistent with the sectoral 2 °C target in refs. [4,5].

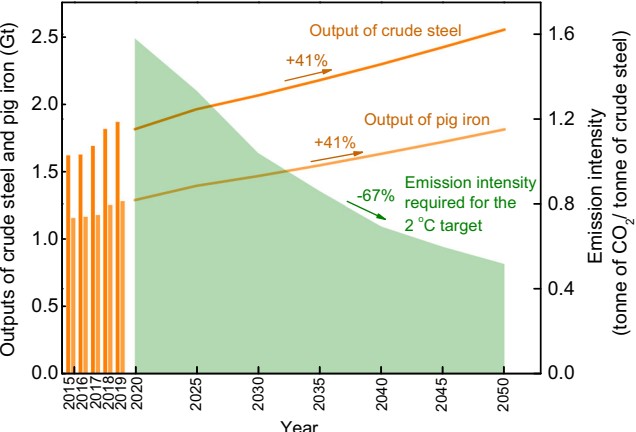

**Fig. 1 Required $CO_2$ emission intensity (tonne of $CO_2$/ tonne of crude steel) with the decreasing budget for the 2 °C climate target versus increasing outputs of crude steel and pig iron globally from 2020 to 2050.** The emission scenario in the iron and steel sector for the 2 °C target (green colour) is set from the International Energy Agency (IEA), which mainly considers short-term energy efficiency improvements and scrap-based electric arc furnace (scrap/EAF) and carbon capture and storage (CCS) in the mid- to long-term[4,5]. The outputs of crude steel and pig iron in 2015–2019 (orange bars) are obtained from the World Steel Association (WSA)[17], while those in 2020–2050 (solid lines) are estimated based on the production scenario given by the United Nations Industrial Development Organization (UNIDO), which mainly considers the strong demand growth in non-OECD countries such as BRICS and ASEAN[5,6]. The future iron and steel outputs could be different from those estimated by UNIDO and therefore, the effect of iron and steel outputs on the potential $CO_2$ emission reduction is further discussed in the potential sensitivity analysis (section: decarbonisation pathways for the iron and steel sector). The $CO_2$ emission intensity is calculated based on sectorial scenarios given by the IEA[4,5] and the production scenario of crude steel and pig iron given by the WSA[17] and UNIDO[5,6].

## Results

**Understanding the waste streams properties to reuse them.** To achieve the 2 °C climate goal proposed in refs. [4,5], the iron and steel sector will need to reduce the emission intensity from 1.58 tonnes of $CO_2$ per tonne of crude steel in 2020 to a target of 0.52 tonnes of $CO_2$ per tonne in 2050 (Fig. 1). In addition, the emission scenario in the iron and steel sector for the 2 °C target is set from the International Energy Agency (IEA)[4,5]; and the outputs of crude steel and pig iron in 2020-2050 are estimated based on the production scenario given by the United Nations Industrial Development Organization (UNIDO)[5,6] (Fig. 1). Based on the annual $CO_2$ emission levels required in 2020-2050 toward the 2 °C climate goal[4,5], we can estimate the $CO_2$ emission reduction potentials of current pathways and schemes based on the heat recovery and resource recycling of waste streams in the iron and steel sector (Methods section).

The BFS forms two final states depending on the cooling rate: glassy and crystalline states. If BFS is in the glassy state, it can be used as a cement feedstock and substituted to limestone calcination $CO_2$ emissions, which has a mean emission intensity of ~0.9 tonnes of $CO_2$ per tonne of cement[32]. If BFS is in the crystalline state, the economic value is significantly reduced[18–20]. Here we assume that the $CO_2$ emissions avoided in the cement production by using waste slag can be claimed by the steel industry, although other accounting methods are open in case cement and steel industries cooperate and integrate their activities to make this reduction happen. The critical cooling rate is the key parameter determining the glass-forming ability of liquid slag, i.e.,

the lowest cooling rate required to fully transform BFS into the glassy state[29–31]. Currently, three main approaches have been proposed to cool hot BFS with heat recovery and resource recycling, namely natural cooling, water quenching, and dry granulation, with quite different practical cooling rates and states of the cooled BFS. Accordingly, three strategies are proposed for BFS treatment (Supplementary note 1): BFS-Glassy/Water, BFS-Glassy/Dry and BFS-Crystalline/Dry.

The crystallisation ability of SS is quite strong due to the high basicity (CaO/SiO₂) and "FeO" concentration, different from BFS. It is difficult to fully avoid the crystallisation behaviour of SS because of the high liquidus temperature. In this case, it will be challenging to obtain a 100% glassy state using a dry granulation method[33,34], as a crystalline state of SS is generally obtained. Nevertheless, if the SS is water quenched and held for a long time, a glassy state with a small crystalline content can still be formed (SS-Glassy/Water). Then, there are two possible approaches for using the remaining unavoidable crystalline SS, i.e., low economic-value recycling for construction, road and landfilling (SS-Crystalline/Dry) and high economic-value reuse of CaO to replace limestone calcination in the cement or steel sector like for slagmaking after necessary iron and phosphorus separations (SS-Crystalline/Dry-R)[18–20,23–25]. Summarising these characteristics, three strategies are proposed for SS treatment (Supplementary note 2): SS-Crystalline/Dry, SS-Crystalline/Dry-R and SS-Glassy/Water.

**Decarbonisation pathways for the iron and steel sector.** Starting from the fundamental properties and treatment strategies for BFS and SS outlined above, we construct nine pathways for the iron and steel sector, consistent with the 2 °C global decarbonisation scenarios and target CO₂ budget in refs. [4,5], named Pathways 1–9 (Table 1). Their potentials for CO₂ emission reduction are calculated based on energy recovery and resource recycling (Methods section). Among all the pathways, Pathways 3 (BFS-Glassy/Water+SS-Crystalline/Dry-R), 4 (BFS-Glassy/Dry+SS-Glassy/Water) and 6 (BFS-Glassy/Dry+SS-Crystalline/Dry-R) have relatively high potentials for CO₂ emission reduction because they allow both energy recovery and resource recycling of BFS and SS, while Pathway 8 (BFS-Crystalline/Dry+SS-Crystalline/Dry) has the lowest potential because it brings only energy recovery (Fig. 2a and Supplementary Figs. 1–3).

From the perspective of CO₂ emission reduction (Table 1), Pathway 6 has the highest potential as it harnesses both resource recycling and energy recovery of BFS and SS. In this pathway, CO₂ emissions could be reduced by 370 and 377 Mt in 2035 and 2050, equivalent to 19.7% and 28.5% CO₂ emission reduction to the 2 °C target requirements in refs. [4,5], respectively. A sensitivity analysis of Pathway 6 (Fig. 2b and Methods section) further shows that 28.5% ± 5.7% CO₂ emission reduction to the 2 °C target requirements in refs. [4,5] can be realised in 2050, considering uncertainties of technological levels in the iron and steel industry, the global crude steel output and related waste streams[6,14,17]. Pathway 5 (BFS-Glassy/Dry+SS-Crystalline/Dry) also shows a relatively high potential, its difference from Pathways 6 being the further utilisation of the cooled crystalline SS, i.e., whether the high content of CaO in SS is effectively recycled, e.g., for cement production or reused for slagmaking in the iron and steel industry. Therefore, in the following analyses, we focus on Pathways 5 and 6 regarding their technological feasibility and economic costs and benefits.

**Technical feasibility.** In Pathways 5 and 6, we assume that the heat in both BFS and SS is recovered, in which BFS is in a glassy state and SS is in a crystalline state. Two methods can be employed to recover the heat in hot slag, namely physical and chemical ones[18–20]. For the physical method, the development of granulation techniques is the main challenge, while chemical gasification methods are only developed as laboratory research with a low technologies readiness level (TRL)[18–20], the selection of granulation and gasification agents is the key issue. Accordingly, five schemes with different TRLs are proposed here to engineer Pathways 5 and 6: physical granulation, air granulation + CO₂ gasification, air granulation + H₂O gasification, CO₂ granulation + CO₂ gasification and CO₂ granulation + CO₂/H₂O gasification, numbered Schemes 1–5, respectively (Supplementary Fig. 20 and Supplementary note 3 and 4).

Based on the mass and energy balances, the yields of final products per tonne of BFS and SS in each scheme and the product yields per tonne of crude steel are calculated (Methods). In Scheme 1, steam, glassy BFS and crystalline SS are the main products, while in Schemes 2–5, syngas composed of CO, H₂, CH₄ and CO₂ is another product (Supplementary Fig. 4 and Supplementary Table 1). The steam could be reused in an existing steam system in the steel industry or used for power generation, while the syngas could be used for power generation or chemical engineering[18–20]. Scheme 1 shows a final energy efficiency of 62%, but Schemes 2–5 produce less steam and more valuable syngas, with the energy efficiencies markedly improved to ~84% due to the direct energy recovery for gasification (Supplementary Fig. 4).

**Table 1 Summary of the nine development pathways based on the inherent potential of waste streams in the iron and steel sector, and their CO₂ emission reductions and the corresponding ratios to the 2 °C target requirements in 2020, 2035 and 2050.**

| Pathway | Key points | Emission reduction (Mt) and ratio to the 2 °C target requirements | | |
|---|---|---|---|---|
| | | In 2020 | In 2035 | In 2050 |
| Pathway 1 | BFS-Glassy/Water + SS-Glassy/Water | 252/8.8% | 268/14.3% | 273/20.7% |
| Pathway 2 | BFS-Glassy/Water + SS-Crystalline/Dry | 191/6.7% | 202/10.8% | 206/15.6% |
| Pathway 3 | BFS-Glassy/Water + SS-Crystalline/Dry-R | 295/10.3% | 313/16.7% | 319/24.1% |
| Pathway 4 | BFS-Glassy/Dry + SS-Glassy/Water | 306/10.7% | 324/17.3% | 331/25.0% |
| Pathway 5* | BFS-Glassy/Dry + SS-Crystalline/Dry | 244/8.5% | 259/13.8% | 264/20.0% |
| Pathway 6* | BFS-Glassy/Dry + SS-Crystalline/Dry-R | 348/12.1% | 370/19.7% | 377/28.5% |
| Pathway 7 | BFS-Crystalline/Dry + SS-Glassy/Water | 165/5.8% | 176/9.4% | 179/13.5% |
| Pathway 8 | BFS-Crystalline/Dry + SS-Crystalline/Dry | 103/3.6% | 110/5.9% | 112/8.5% |
| Pathway 9 | BFS-Crystalline/Dry + SS-Crystalline/Dry-R | 208/7.3% | 221/11.8% | 225/17.0% |

These pathways are summarised according to fundamental properties of waste streams including blast furnace slag (BFS) and steel slag (SS), and technological levels. The CO₂ emission reductions and the corresponding ratios to the 2 °C target requirements in refs. [4,5] by these pathways are calculated based on the energy recovery and resource recycling of these waste streams. The differences between Pathways 2 and 3, 5 and 6, and 8 and 9 results from the further utilisation of cooled crystalline SS (SS-Crystalline/Dry or SS-Crystalline/Dry-R), i.e., whether the high concentration of CaO in the cooled crystalline SS could be recycled after necessary phase separation considering the technological advancements in the mid- to long-term. In this study, the technological schemes applied to engineer Pathways 5 and 6 (marked as *) are further discussed due to their high-potential CO₂ emission reductions for the 2 °C climate target.

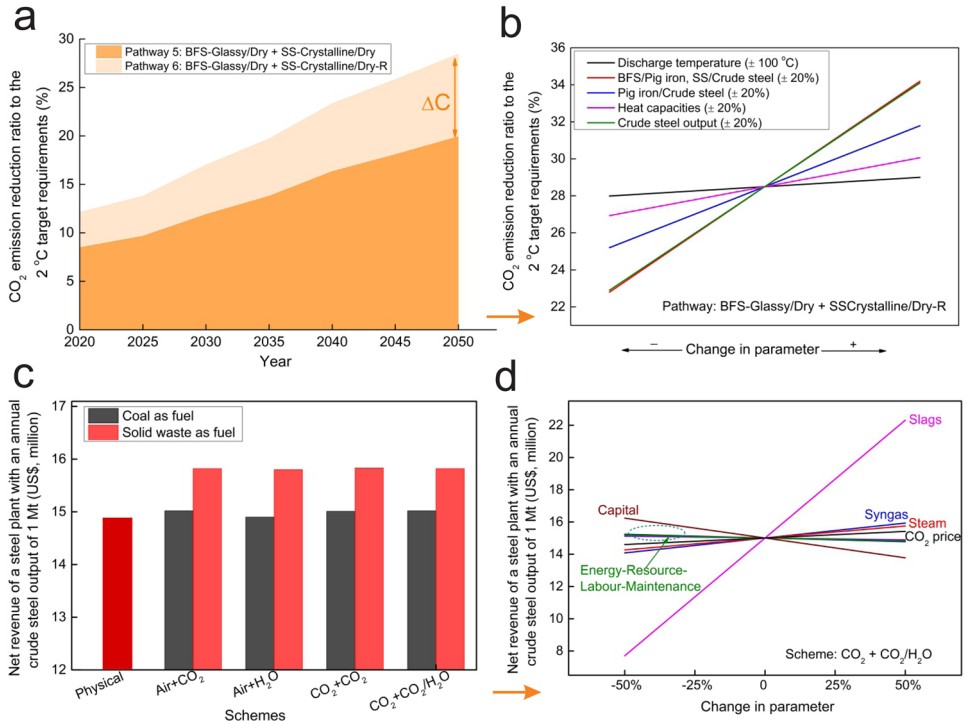

**Fig. 2 Pathways to the sectoral 2 °C climate target in the iron and steel industry and economic analyses of the five technological schemes applied to engineer the high-potential pathways. a** $CO_2$ emission reduction ratios to the 2 °C target requirements in refs. [4,5] in the iron and steel industry by the two high-potential pathways, Pathway 5 (BFS-Glassy/Dry+SS-Crystalline/Dry, dark orange colour) and Pathway 6 (BFS-Glassy/Dry+SS-Crystalline/Dry-R, light orange colour) (Supplementary Figs. 1–3 for other pathways). BFS and SS represent blast furnace slag and steel slag, respectively. The potential difference ($\Delta$C) results from the further utilisation of cooled crystalline SS, i.e., whether the high concentration of CaO in cooled crystalline SS could be recycled after necessary phase separations considering the technological advancements in the mid- to long-term. **b** The potential sensitivity of the highest-potential Pathway 6 in 2050 in terms of $CO_2$ emission reduction ratio to the 2 °C target requirements in refs. [4,5]. **c** Net revenue of a steel plant with an annual crude steel output of 1 Mt by the five technological schemes, outlined in Supplementary Fig. 20, applied to engineer Pathways 5 and 6 (black and red colours represent the use of coal and solid waste as fuel, respectively). **d** Economic sensitivity of this steel plant by the technological scheme of $CO_2$ granulation + $CO_2/H_2O$ gasification (Supplementary Figs. 15–18 for other technological schemes). The results are first calculated at the levels of per tonne of glassy BFS and of crystalline SS, and then the results at the levels of per tonne of crude steel and of a steel plant with an annual crude steel output of 1 Mt are further derived using BFS/Pig iron, SS/Crude steel and Pig iron/Crude steel ratios of 0.26, 0.13 and 0.71, respectively (Supplementary Fig. 19).

**Economics of the steel sector.** We perform a cost-benefit analysis of each scheme based on process costs incurred from investment and operation, and a $CO_2$ price assumed at a certain level (30 US $/tonne). The two benefits are the products generated by the improved industrial processes and the revenues from avoided emissions at the $CO_2$ price fixed (Supplementary note 5). The operating costs are divided into maintenance, labour, resource and energy costs, while valuable products include BFS, SS, steam and syngas. For the methods of chemical gasification (Schemes 2–5), two fuels can be employed, namely coal and solid wastes such as biomass and sludge, and here, we consider both with different material costs and fossil $CO_2$ emissions. Based on our economic analyses per tonne of BFS and SS (Supplementary Figs. 5–14), we estimate the net economic balance of a steel plant with an annual crude steel output of 1 Mt and at the global level (Methods section).

Figure 2c shows that, ~US$14.9 million will be generated annually for a typical 1 Mt steel plant by Scheme 1. Under Schemes 2–5, the annual revenue slightly increases if coal is used as the gasification fuel and the revenue increases to US $15.8 million if solid wastes are utilised as the fuel. At the global level, if all plants adopt the proposed technologies (Supplementary Table 2), the net revenues of these proposed schemes will increase to a large scale. By 2035, approximately US$32 billion could be generated by Scheme 1 and US$35 billion by

Schemes 2–5 employing solid wastes as the fuel. By 2050, approximately US$38 billion is estimated to be generated by Scheme 1, while this revenue further increases to US$40 billion from the universal adoption of Schemes 2–5.

Economic uncertainty originates from the varying product price, process and carbon price, as estimated by sensitivity studies. Figure 2d (Supplementary Figs. 15–18) shows that for all schemes, the net revenue is mostly sensitive to the slag price because cooled slag (glassy BFS in particular) accounts for the dominant valuable product. If the slag price increases by 50%, the revenue increases from US$15 to US$22 million, while if the slag price decreases by 50%, the total revenue drops to US$8 million. As a result, the formation of BFS in the glassy state appears to be a key target for BFS treatment for all the proposed schemes. Comparatively, the effects of labour, maintenance and energy costs are smaller than the revenues from better reuse of slag products in our analysis. We acknowledge uncertainties in estimating labour and maintenance for slag recovery technologies, as these solutions are not implemented at scale today. In Scheme 1, the capital cost, steam price and assumed $CO_2$ price play comparable roles. In Schemes 2–5, the syngas price also remarkably affects the plant economics as syngas is another valuable product, while the effect of the gasification fuel price is limited. That could be further calculated at the global level based on the economic sensitivity at the level of a steel plant since they show the same sensitivity variables and ratios.

**Integration with CCS to contribute to deeper decarbonisation.** As shown in the analysis of emission reduction potentials from heat and materials reuse (Table 1), up to 28.5% $CO_2$ emission reduction to the sectoral 2 °C target requirements in refs. [4,5] can be realised by the best pathway, Pathway 6. This target requires the coupling of CCS to the iron and steel production in different steps such as in mainstreams, and new low carbon production technologies like TRTs and CCPPs to finally reduce the emission intensity to the target of 0.52 tonnes of $CO_2$ per tonne of crude steel in 2050[4–6].

Based on the global adoption of reuse technologies of Table 1, the iron and steel industry could generate additional revenues of US$35 ± 16 and US$40 ± 18 billion globally in 2035 and 2050, respectively. These revenues could be invested to further reduce the $CO_2$ emissions by CCS. We consider CCS integrated with Pathway 5 or 6, at an average CCS unit cost of US$30 per tonne of $CO_2$ avoided[5,6,35]. The integration of CCS and the reuse of waste streams could be realised from the material flow: after phase separation and post-treatment, the recovered calcium-based materials extracted from BFS and SS could be used for CCS[36]. Employing CCS will reduce an additional 0.5 tonnes of $CO_2$ per tonne of crude steel. Figure 3 suggests that strategies involving CCS could lead to equivalent $CO_2$ emission reduction to the carbon budget in refs. [4,5] in 2042 for Pathway 6 + CCS, and in 2045 for Pathway 5 + CCS, respectively. Here we have shown ambitious figures, with right policy support and assuming no barrier to scalability, technical progress allowing industrialisation, and effectiveness of CCS attached to steel facilities.

The achievability of our proposed pathways and schemes are determined by technological improvements above current levels. For the BFS treatment, the main technological challenge is to obtain a glassy state using a dry granulation method. This target is consistent with current technological developments for cooling hot BFS where extensive granulation processes have been developed[18–20]; for example, the concentrations of $Al_2O_3$[29] and CaO/$SiO_2$[30,31] in BFS have been modified. For the SS treatment, recycling crystalline SS is the main challenge. The main valuable components include CaO, FeO, $P_2O_5$, etc., and their recycling relies on effective separation to enrich them in different phases. The main principles are to control the phase transformations during the cooling process by modifying the temperature schedule, atmosphere and material additions[33,37]. For the reuse of slag in the iron and steel industry like for slagmaking, the final chemical compositions should be well controlled by necessary phase separations and impurity removals[23–25]. Recently, other applications of SS such as agricultural fertilisers[26,38] and soil improvement agents[39] have also been developed.

Another important barrier affecting the $CO_2$ emission reductions of the proposed pathways and schemes is the deep integration of waste stream treatment with CCS. Here we discuss two main flows, namely capital flow where CCS implementation is funded by the revenue generated by waste stream treatment, and material flow where CaO used for CCS is produced from the waste streams. As CCS facilities are configured for different sites of ironmaking and steelmaking processes, the average costs can be quite different[5,6,35]. It is expected that the potential for $CO_2$ emission reduction by these pathways and schemes will further expand with technological progress in CCS in the mid- to long-term[5,6,35]. Moreover, the pathways and technology schemes discussed in this study could be an important wedge for approaching net-zero emissions for the steel industry as proposed by the EU Green Deal[11], along with other transformational changes such as hydrogen- and green electricity-based metallurgy and smart carbon usage[11–13].

In summary, we explore the feasibility of decarbonising the iron and steel industry for the sectoral 2 °C climate target by integrating the inherent potential of waste streams with internal funding of carbon capture and storage. In a pathway where the output of blast furnace slag is in a glassy state and the output of steel slag in a crystalline state, both cooled by a dry agent, a

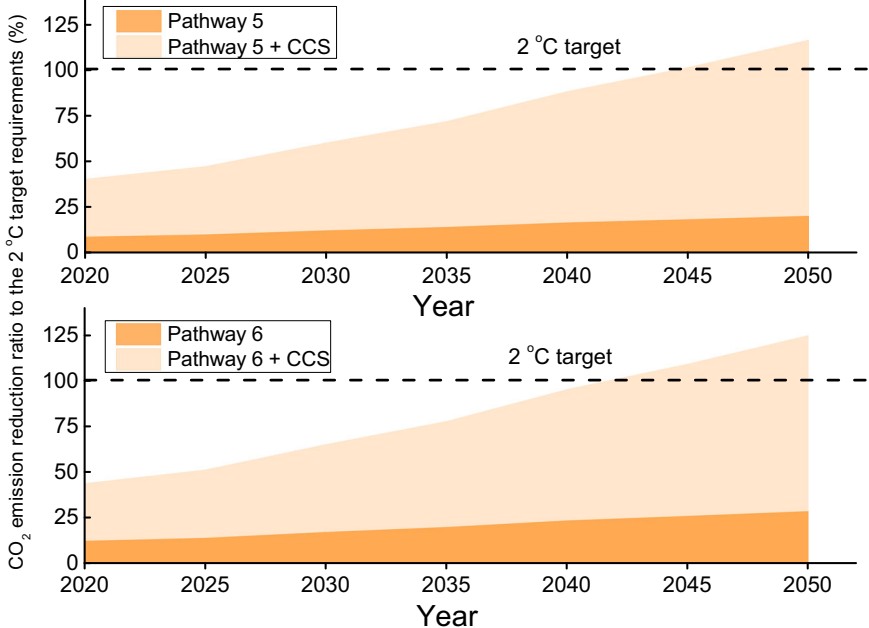

**Fig. 3 $CO_2$ emission reduction ratios to the 2 °C target requirements in the iron and steel sector based on the inherent potential of waste streams integrated with internally funded carbon capture and storage.** By Pathways 5 (BFS-Glassy/Dry+SS-Crystalline/Dry) and 6 (BFS-Glassy/Dry+SS-Crystalline/Dry-R) alone, 20.0% and 28.5% $CO_2$ emission reductions to the 2 °C target requirements in refs. [4,5] in the iron and steel sector can be realised in 2050 (light orange colour). BFS and SS represent blast furnace slag and steel slag, respectively. When Pathways 5 and 6 are deeply integrated with CCS (carbon capture and storage) based on capital and material flows, equivalent $CO_2$ emission reduction to the 2 °C target requirements in refs. [4,5] can be realised before 2050 (2045 and 2042 based on Pathway 5 + CCS and Pathway 6 + CCS, respectively) (dark orange colour).

substantial revenue (US$15 ± 7 million) will be yielded annually for a steel plant with an annual crude steel output of 1 million tonnes. If this revenue is used to fund carbon capture and storage, equivalent $CO_2$ emission to the sectoral 2 °C target requirements is expected to be reduced in the iron and steel industry before 2050 without any external investments, therefore leading to the realisation of the long-term decarbonisation target and sustainable development of this emission-intensive industry.

## Methods

**Estimation of $CO_2$ emission reduction potential in various pathways.** To estimate the potential of $CO_2$ emission reductions in different pathways in 2020–2050, the global outputs of crude steel should be first predicted (Supplementary Table 3). Here, we use the production scenario estimated by the United Nations Industrial Development Organization (UNIDO)[5,6]. To further estimate the output of pig iron, the weight ratio of pig iron to crude steel (Pig iron/Crude steel, named as α) should be determined. Based on the outputs of crude steel and pig iron between 2008 and 2018, this parameter is in the range of 0.71–0.75 considering the two main processes of scrap-based electric arc furnace (scrap/EAF) and blast furnace/basic oxygen furnace (BF/BOF) in the iron and steel industry[9,14,17] (Supplementary Table 3). Due to technological progress and the increasing utilisation of scrap/EAF process, this ratio could be slightly decreased in the future[9,14,17]. Therefore, in this study, the Pig iron/Crude steel ratio is selected to be 0.71, based on which the global output of pig iron in 2020–2050 is estimated (Supplementary Table 3).

After the global outputs of pig iron and crude steel are obtained, two other parameters are required to obtain the global productions of BFS and SS, namely the weight ratio of BFS to pig iron (BFS/Pig iron, named as β) and that of SS to crude steel (SS/Crude steel, named as γ). At the current global level in 2020, these two parameters are approximately 0.26 and 0.13, respectively[14–17]. Due to technological progress, the discharge of BFS and SS could decrease continuously[15,16]. Therefore, it is assumed that the BFS/Pig iron and SS/Crude steel ratios will decrease at rates of 0.002/year and 0.001/year from 2020 to 2050, respectively[14–17]. Based on these ratios, the global productions of BFS and SS from 2020 to 2050 can be estimated (Supplementary Table 3).

In addition to the outputs of BFS and SS, other significant properties determining the potential of the present pathways include the discharge temperature and heat capacities of BFS and SS. Regarding the discharge temperature, BFS and SS are generally discharged at temperatures as high as 1550 °C and 1600 °C, respectively. With the further advancement of ironmaking and steelmaking technologies, these temperature points will vary in the future; for example, a lower discharge temperature is expected to decrease the energy consumption in the iron and steel sector. This will affect the thermal energy carried by the hot slag. There are two methods to obtain the heat capacities of BFS and SS (Supplementary note 6) based on the basic principles of mass and energy balances. Based on the heat capacities and discharge temperatures of BFS and SS, the thermal heat carried by per kg of slag ($E_s$) is calculated by means of Eq. (1).

$$E_s = \int_{T_0}^{T_1} Cp(T)dT = \sum_i Cp_i \Delta T_i \qquad (1)$$

where $E_s$ is the thermal heat carried per kg of slag; $Cp(T)$ is the heat capacity at temperature $T$; $T_0$ is room temperature 25 °C and $T_1$ is the discharge temperature of the slag, where $T_1$ is equal to 1550 °C for BFS and 1600 °C for SS.

Furthermore, based on the thermal heat carried by per kg of BFS and SS and the annual outputs, the total global energy potential in the iron and steel sector can be calculated by means of Eq. (2). Here the calculation process is simplified where the heat capacities of BFS and SS are selected to be constant at 1.15 kJ/kg/K and 1.05 kJ/kg/K, respectively, which are in the medium range of the heat capacities (Supplementary Table 4). After the potential energy recovery from BFS and SS is obtained, the corresponding potential for $CO_2$ emission reduction based on the energy recovery from the high-temperature slag can be calculated as follows based on the emission intensity of standard coal, $9.03 \times 10^{-8}$ kg $CO_2$/J (210 pounds/MMBtu)[40].

$$E_t = E_{BFS}O_{BFS} + E_{SS}O_{SS} \qquad (2)$$

where $E_t$ represents the total energy potential in the iron and steel sector; $E_{BFS}$ and $E_{ss}$ represent the thermal heat carried per kg of BFS and SS, respectively; and $O_{BFS}$ and $O_{ss}$ represent the annual outputs of BFS and SS, respectively.

In addition to energy recovery, another potential route of $CO_2$ emission reduction from the utilisation of BFS and SS is based on resource recycling. Generally, after cooling, the slag can present two states, namely glassy and crystalline states. The critical cooling rate is usually used to evaluate the crystallisation ability of a high-temperature liquid slag[29–31]; i.e., if the process cooling rate is larger than the critical cooling rate, such as that of quenching by water, the liquid slag will be transformed into a glassy state due to the limited time for structural relaxation. In contrast, if the cooling rate is smaller than this value, such as that of naturally cooling, the liquid slag will be transformed into a crystalline state composed of various minerals. Slag in a glassy state, with high

hydraulic activity, can be used as raw materials for cement manufacturing considering the huge consumption of cement accompanied with the expanding urbanisation and industrialisation globally, which accounts for the dominant resource recycling of BFS and SS[18–20]. The resource recycling potential of BFS and SS is mainly based on the use of CaO as raw materials for cement production. It is assumed that once the BFS and SS are used for cement manufacturing, the corresponding amount of $CO_2$ will be reduced based on the calcination reaction of $CaCO_3$, described by means of Eq. (3). Based on the chemical compositions of BFS and SS[14,18–20] (Supplementary Table 4), in this study, the CaO contents in BFS and SS are assumed to be 42 wt. % to simplify the calculations.

$$CaCO_3 = CaO + CO_2, \Delta H_r^{900°C} = 166.6 \text{kJ/mol} \qquad (3)$$

In addition to direct CaO replacement, there are two other types of energy savings and emission reductions once the glassy slag is recycled for cement manufacturing. The first is the reaction heat of Eq. (3) since it is an endothermic reaction, i.e., once 1.00 mole of CaO carried by slag is used for cement manufacturing at 900 °C, 166.6 kJ energy will be saved. Based on the emission intensity of standard coal of $9.03 \times 10^{-8}$ kg $CO_2$/J (210 pounds/MMBtu)[40], it can be calculated that equivalently 0.34 moles of additional $CO_2$ will be reduced once 1.00 mole of CaO is replaced by the CaO-containing slag. The second is the utilisation of other components in slag, such as $Al_2O_3$ and $SiO_2$, for cement production because they are also the basic components in cement materials[18–20]. In the present study, this last type is ignored since its effect is much smaller than that of $CO_2$ emission reduction from the replacement of $CaCO_3$ calcination. Therefore, from the perspective of resource recycling, a total of 1.34 moles of $CO_2$ emissions will be reduced once 1.00 mole of CaO is replaced by the corresponding amount of CaO in slag.

Based on the energy recovery and resource recycling of BFS and SS, the potential of $CO_2$ emission reduction by the individual pathways is first estimated. Next, the potential emission reduction ratios to the 2 °C climate target requirements in refs. [4,5] by these pathways are calculated based on the emission reductions through these pathways and the individual required $CO_2$ emission intensities in each year from 2020 to 2050 under the 2 °C emission scenarios in refs. [4,5] and the steel production scenarios in refs. [5,6]. There are three methods to define the emission reduction ratios. Firstly, the annual emission reductions by the proposed pathways are divided by the current emissions annually to obtain a contribution ratio. However, the emission reductions caused by other methods like high-cost CCS are ignored. Secondly, the annual emission reductions are divided by the annual emission budget in refs. [4,5] to obtain a contribution ratio. This is a deeper emission reduction based on the 2 °C climate target to achieve a zero-emission goal. Thirdly, the annual emission reductions are divided by the gap between the current emission level and the level under the 2 °C climate target. However, it is difficult to define the contribution level at the early stage of 2020–2050 because the emission potential by the proposed pathways greatly exceeds the gaps. Therefore, in this study, we use the second method to define the emission reduction ratios by the various pathways and schemes.

From the estimation of the energy recovery and resource recycling potentials of BFS and SS, it can be observed that the emission reductions are influenced by the production of BFS and SS and their heat capacities and discharge temperatures; the former is further determined by the crude steel output and the Pig iron/Crude steel, BFS/Pig iron and SS/Crude steel ratios. As these factors vary, the potential of various scenarios and their emission reductions to the 2 °C target requirements will correspondingly change, especially considering technological advancements such as the increasing use of scrap/EAF and the uncertainties of crude steel demand in the future due to interventions in different countries and areas and global carbon price[6,14–17]. Therefore, the sensitivity of Pathway 6 is discussed, in terms of $CO_2$ emission reduction ratio to the 2 °C target requirements.

Several assumptions and simplifications are made. First, regarding the discharge temperature, it is assumed that it will vary by ±100 °C due to the operational changes in ironmaking and steelmaking processes, i.e., the discharge temperature of BFS varies from 1450 to 1650 °C and that of SS varies from 1500 to 1700 °C. Second, regarding the Pig iron/Crude steel, BFS/Pig iron and SS/Crude steel ratios, we assume that they will vary by ±20% due to varying operational conditions such as technological advancements in ironmaking and steelmaking and the degradation of iron ore. Additionally, to simplify the analyses and discussions, we assume that the two ratios, BFS/Pig iron and SS/Crude steel, will change in the same direction, increasing or decreasing simultaneously. Third, regarding the heat capacities, we assume that they will vary by ±20%, due to the varying chemical compositions of BFS and SS.

**Equilibrium calculation of the gasification process.** In this study, two types of methods regarding the energy recovery from BFS and SS are considered, namely physical granulation and chemical gasification methods. For the gasification methods currently of low TRL[16–18], three types of agents can be used, namely $CO_2$, $H_2O$ (steam) and a mixture of $CO_2$ and $H_2O$, and other factors determining the results are the gasification fuel, gasification temperature, and fuel/agent ratio. The equilibrium calculation is first performed using FactSage software[41], which directly determines the further process analysis results of Schemes 2–5. Here, the methodology for the equilibrium calculations is briefly highlighted (Supplementary note 7). First, two kinds of fuel can be selected, namely coal and solid wastes such as biomass and sludge; both are considered here, associated with the final economic results. The typical compositions of the fuel are selected based on previous literature[42–44] (Supplementary Table 4), where only C, H

and O elements are considered, and the presence of ash is ignored to simplify the analyses. We choose four parameters to assess the gasification results, including CO content ($\eta_{CO}$), $H_2$ content ($\eta_{H_2}$), carbon efficiency (CE) and hydrogen efficiency (HE), defined as follows:

$$\eta_{CO} = \frac{V_{CO}}{Vs} \times 100\% \tag{4}$$

$$\eta_{H_2} = \frac{V_{H_2}}{Vs} \times 100\% \tag{5}$$

$$CE = \frac{n_{C,CO} + n_{C,CH_4}}{n_{C,fuel}} \times 100\% \tag{6}$$

$$HE = \frac{n_{H,H_2} + n_{H,CH_4}}{n_{H,fuel}} \times 100\% \tag{7}$$

where $Vs$, $V_{CO}$, $V_{H_2}$, $V_{CH_4}$ and $V_{CO_2}$ represent the volumes of total syngas, CO, $H_2$, $CH_4$ and $CO_2$, respectively; $\eta_{CO}$ and $\eta_{H_2}$ represent the contents of CO and $H_2$ in syngas, respectively; CE and HE represent the carbon efficiency and hydrogen efficiency during gasification, respectively; $n_{C,CO}$, $n_{C,CH_4}$ and $n_{C,fuel}$ represent the moles of carbon in CO, $H_2$ and gasification fuel, respectively; and $n_{H,CO}$, $n_{H,CH_4}$ and $n_{H,fuel}$ represent the moles of hydrogen in CO, $H_2$ and fuel, respectively.

The selection of gasification conditions is based on the four efficiencies in the equilibrium calculation results (Supplementary note 8). For $CO_2$ gasification, to confirm the full gasification of fuel and the high gasification efficiencies, and to ensure a low cost of syngas separation after gasification, a $CO_2$/fuel ratio of 2:1 is employed. Similarly, to simplify the calculations and to compare the results with those under $CO_2$ gasification, the same $H_2O$/fuel ratio of 2:1 is used for $H_2O$ gasification. For gasification using a mixing agent of $CO_2$ and $H_2O$, a $CO_2$/$H_2O$ mole ratio of 1:1 is employed, and as a result, a final mole ratio of fuel/$CO_2$/$H_2O$ of 1:1:1 is selected for further process analysis.

**Process analysis of various schemes**. The energy and resource analyses of the five schemes are mainly performed based on the principles of energy and mass balances, and here, only the key equations for the individual steps of each scheme are discussed. For Scheme 1, three steps make up the whole process including air-slag granulation, air-slag heat transfer and air-steam heat transfer, and the main equations characterising each step are shown as follows (Supplementary note 4):

Step 1:

$$m_{BFS} \times C_{p,BFS}(T_{BFS,2} - T_{BFS,1}) = m_{a,1} \times C_{p,a} \times (T_{a,2} - T_{a,1}) \tag{8}$$

where $m_{BFS}$ and $m_{a,1}$ represent the masses of BFS and air, respectively; $C_{p,BFS}$ and $C_{p,a}$ represent the heat capacities of BFS and air, respectively; and $T_{BFS,1}$, $T_{BFS,2}$ and $T_{a,1}$, $T_{a,2}$ represent the temperature points of BFS and steam before and after heat transfer, respectively.

Step 2:

$$m_{BFS} \times C_{p,BFS}(T_{BFS,3} - T_{BFS,2}) = m_{a,2} \times C_{p,a} \times (T_{a,2} - T_{a,1}) \tag{9}$$

where $m_{BFS}$ and $m_{a,2}$ represent the masses of BFS and air, respectively; $C_{p,BFS}$ and $C_{p,a}$ represent the heat capacities of BFS and air, respectively; and $T_{BFS,2}$, $T_{BFS,3}$ and $T_{a,1}$, $T_{a,2}$ represent the temperature points of BFS and steam before and after heat transfer, respectively.

Step 3:

$$(m_{a,1} + m_{a,2}) \times C_{p,a} \times (T_{a,2} - T_{a,3}) = m_s \times [C_{p1,s}(T_{s,2} - T_{s,1}) + \Delta H_s + C_{p2,s}(T_{s,3} - T_{s,2})] \tag{10}$$

where $m_{a,1} + m_{a,2}$ and $m_s$ represent the masses of hot air and steam, respectively; $C_{p,a}$, $C_{p1,s}$ and $C_{p2,s}$ represent the heat capacities of hot air and water at 25–100 °C and steam at 100–200 °C, respectively; $\Delta H_s$ represents the latent heat of steam at 100 °C; and $T_{a,2}$, $T_{a,3}$ and $T_{s,1}$, $T_{s,2}$, $T_{s,3}$ represent the temperature points of air and steam, respectively.

For Schemes 2–5, the energy balance of the whole process, mainly composed of the granulation and gasification steps, is expressed by the following equation, where the calculations and the related simplifications of each part regarding the energy balance are more complex (Supplementary note 4):

$$Q_{fuel,s} + Q_{fuel,l} + Q_{agent,s} + Q_{agent,l} + Q_{BFS} = Q_{loss} + Q_{syngas,s} + Q_{syngas,l} + Q_{steam} \tag{11}$$

where $Q_{fuel,s}$ and $Q_{fuel,l}$ represent the sensible and latent heat of the fuel, respectively; $Q_{agent,s}$ and $Q_{agent,l}$ represent the sensible and latent heat of the gasification agent, respectively; $Q_{BFS}$ represents the thermal heat in the BFS; $Q_{loss}$ represents the heat loss in various steps; $Q_{syngas,s}$ and $Q_{syngas,l}$ represent the sensible and latent heat of the syngas, respectively; and $Q_{steam}$ represents the thermal energy in the steam product.

Based on the energy and mass balance analyses, the product yields per tonne of BFS and SS through the different schemes can be obtained, and accordingly, we can further derive the product yields per tonne of crude steel of a steel plant with an annual crude steel output of 1 Mt (Supplementary Fig. 19), where the BFS/Pig iron, SS/Crude steel and Pig iron/Crude steel ratios are selected to be 0.26, 0.13 and 0.71, respectively.

**Economic and sensitivity analysis**. Based on the process analysis results, as well as the prices of each input factor and product, the costs and benefits of different schemes can be estimated (Supplementary note 5). Here, only the key issues are summarised. First, because the prices of input factors and products, obtained from the pilot-scale trials and market prices, are quite different in various regions, they are harmonised considering global income and labour levels[44–46]. Second, based on the energy balance, the prices of steam and syngas are transformed into the corresponding weight of natural gas with a constant price. The prices of the individual input factors and products after harmonisation and the related conversion parameters are summarised in Supplementary Table 5,[12,44–49]. Third, based on the process analysis results, economic analysis can be performed from four levels (Supplementary Fig. 19), namely per tonne of BFS and SS (Supplementary Tables 6–10), per tonne of crude steel (Supplementary Table 11), a steel plant with the annual crude steel output of 1 Mt (Supplementary Table 12), and the global iron and steel sector (Supplementary Table 2). In this study, only the global level and the level of a steel plant are discussed in detail. The relationship between the net revenue (NR) per tonne of crude steel and those per tonne of BFS and SS is shown as follows:

$$NR = \alpha \big[ (B_{BFS1} + B_{BFS2} + \dots + B_{BFSn}) - (C_{BFS1} + C_{BFS2} + \dots + C_{BFSn}) \big] + \beta\gamma \big[ (B_{SS1} + B_{SS2} + \dots + B_{SSn}) - (C_{SS1} + C_{SS2} + \dots + C_{SSn}) \big] \tag{12}$$

where $\alpha$, $\beta$ and $\gamma$ are the BFS to pig iron, the SS to crude steel and the pig iron to crude steel ratios, respectively; $B_{BFS}$ and $C_{BFS}$ are the different benefits and costs per tonne of BFS during the treatments (same for the SS subscript).

Regarding the economics of a plant, for the various schemes, the market situation will change with the varying operational conditions in the mid- to long-term such as the material prices, labour cost and $CO_2$ price. Here the $CO_2$ price is assumed to be US$30 per tonne of $CO_2$, which is in consistence the $CO_2$ avoided cost by CCS[5,6,35,50]. This will influence the costs and benefits of these schemes and the net revenues, and therefore, it is necessary to clarify the effects of varying factors on the economics of a steel plant. Here we assume that the factors will vary by 0 to ±50%, which represents the economic stability of a steel plant in overcoming market changes. Similarly, based on the economics of the level per tonne of BFS and SS, the economic sensitivity at this level can be firstly investigated. Based on the economic sensitivity per tonne of BFS and SS, the economic sensitivity per tonne of crude steel can be calculated, where the BFS/Pig iron, SS/Crude steel and Pig iron/Crude steel ratios are also selected to be 0.26, 0.13 and 0.71, respectively. Accordingly, the economic sensitivity at the plant level is finally analyzed for an annual crude steel output of 1 Mt (detailed in the section of "Economics of the steel sector").

## Data availability

The data that support the findings detailed in this study are available in the paper and its Supplementary Information.

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

## Acknowledgements
Supports by the National Natural Science Foundation of China (51522401, 51772141 to Zt.Z, 72173133 to J.M., and 42071022 to Zz.Z) and Natural Environment Research Council (NE/V002414/1 to J.M.) are acknowledged. Additional support was provided by Guangdong Province Universities and Colleges Pearl River Scholar Funded Scheme 2018 (2018 to Zt.Z).

## Author contributions
Y.S. and Zt.Z. conceived the idea for the project. Y.S. and S.T. collected the data. Zt.Z., Zz.Z. and J.M. guided this study. Y.S. built the research methodology and performed all calculations. P.C. gave important guidance on the scenario selection and analyses. Y.S., S.T., P.C., Zz.Z., J.M. and Zt.Z. discussed the results and contributed to writing the paper.

## Competing interests
The authors declare no competing interests.
