## [Peer Review File · Nature Communications]

Peer Review File, first reviewer comments –

Reviewer #1 (Remarks to the Author):

The paper addresses a relevant issue related to the GHG emissions of the steel sector and to mitigation solutions to achieve the Paris Agreement target (2°C).

It is very detailed and complex and, as such, is rather difficult to read by an expert in the field and probably confusing for readers from outside the field.

The difficulty stems from the format of "article, appendices, supplementary materials comments, tables and figures", which are embedded into each other like Russian dolls and prevent a continuous reading of the paper's argument and claims. I understand that this fragmentation of the rhetoric argumentation of the paper is a requirement of the journal, not something that the authors particularly want to stick to.

The paper states in its conclusions that controlling the various kinds of slag produced in a steel mill in terms of energy and material recovery can provide enough revenue to pay for abatement technologies based on CCS and thus make it possible to achieve the 2° target. Not simply for a steel mill but for the whole steel sector, worldwide.

This is a rather extraordinary claim. It exhibits a solution of which the steel sector is not aware and thus has a kind of magic character!

Now, let us analyze more in depth what we have understood of this complex paper and thus try to express an opinion on the value of this claim.

The complexity of the paper stems from the fact that it examines a large set of technology-options for handling slag (BFS and SS), carries out cost analyses and does it for a time extension from 2020 to 2050 (foresight exercise).

The puzzling part of the paper is that focusing on a tiny part of steel production operations is enough to solve an otherwise difficult and forbearing problem (GHG mitigation).

Two technology pathways are examined: recovery of energy (heat) from hot slag and material recovery from the slag.

Heat recovery has been a research and development theme in the steel sector, in Japan and Europe, for very many years and the search continues for practical solutions that can be implemented: many papers and grey literature reports were published but little operational feedback as the technologies are not yet mature enough for plant operative to invest in them and their economic benefits did not materialize. Therefore, the economic benefits that you show in the paper are difficult to match with what is actually going on in the real world. You probably overestimate the yield of energy recovery and its quality? And put too much expectation in the technologies? Thus the paper makes a long list of options (your pathways, which are fine), attaches a series of technologies to each one them and this produces a complex set of bullet points which describe options, some of them far-fetched, and most not necessarily feasible.

Regarding the recovery of material, mostly "calcium" for its hydraulic properties, we certainly agree. However, it seems to me that BF slag is 100% recovered and already sold to the cement industry in important parts of the world, in Europe for example. BOF slag is recovered for lesser applications, like roadmap and similar uses, which is paid less economically and environmentally, but it is recovery and more ends up in slag heaps. EAF slag recovery is even less. The reason for the lower status of Steelmaking slag is its composition, which does not match the composition requirements of cement making: in this case, composition needs to be amended by using high-temperature processing which has been called slagmaking... a technique, which, anyway, remains confidential.

Therefore, at the end of the day, "Ca recovery" cannot be presented as a widely open option for the steel sector, as it is already in place in important regions of the world, where steel technology is quite advanced.

Last but not least, selling BF slag as clinker additive to the cement sector did not generate enough revenue to make CCS possible! Extending the technique to more steel capacity in the world looks therefore more like technology transfer than the implementation of new breakthrough technology.

The list of references is long and impressive, but centered either on east Asia papers (I cannot tell if there are many self-references or not) or on review papers from western papers. Moreover, there are clear gaps in the review of European papers and documents. And it is not clear what refers to operating practices in the steel sector and academic papers presenting models and schemes of technological solutions and principles, although the emphasis is on the latter rather than on the former one. Moreover you seem to be cherry-picking data among the various references.

After these general comments, more detailed questions are listed in what follows.

1. "9% of annual global anthropogenic CO₂ emissions": there is an emission range from 4% to 9%, depending on reference (anthropogenic emissions without or with ILUC) and level of specific emissions. Your references 1,2 are not very convincing!
2. "1.58 tons of CO₂ per ton of crude steel in 2020": this is a low-value. Actual data (see the paper that you call UNIDO in your references) are much more dispersed and exhibit a higher average. It depends on the boundaries inside which emissions are accounted (e.g. are the coke ovens in or out? Etc.).
3. The list of low-carbon new technology listed lines 54 and fol. Is a bit strange: TRT are very common in Asia, where electricity is expensive, start to appear in Europe; CCPPs are not relevant here (by the way, are power plants inside "the boundaries" or not); and COREX definitely does not reduce GHG emissions, quite the contrary, and, anyway, there are so few of them in the world...
4. Line 65: large internal potential is not quite true, at least where advanced world class steel mills are concerned; "valorized" is an open question, as the reason why so many technologies have been experimented upon is that they did not provide enough revenue (ROI) to be implemented industrially.
5. Line 71: this is already practiced and for a very long time (> 50 years?). On the other hand, to which industry should the reduction of emissions be allocated? Steel or cement? The issue is unresolved, as the cement industry is not willing to yield: compare the price of cement to the price of CO₂!
6. Line 74: where do these estimates (potential emission reduction) come from? As a matter of fact, I did not understand how you extrapolated costs to 2030 and 2050. For example, you assume the cost of CO₂ to be 30€/t today. How high will they be in 2050? They'll remain at 30 or rise to 500, as some literature work suggest?
7. Line 86: it sounds like a good idea to invest the revenue (if there is any!) of low-carb technologies to carbon mitigation. However, this is not necessarily how a steel company operates. Usually revenues are not tagged to any specific expenditures.
8. Line 107: long-time held \diamond held for a long time?
9. Line 130: a consequential approach is missing in the argument. If slag is added to clinker, then it may replace Portland cement but also add to overall cement production and thus not substitute anything, cf. the well-known Braess's paradox in road traffic modeling, whereby when more lanes are added to a freeway, traffic increases rather than became smoother.
10. As pointed out before, blast furnace and steelmaking slags are quite different.
11. Line 196: "The labor, maintenance and energy costs all have limited influences on the economics of a steel plant": this is what your graphs shows but it is very counterintuitive. On the other hand, the results are very sensitive to the price of by-products, which is also counter intuitive.
12. Ligne 210, Corex et les autres procédés sont déjà cités.
13. CCS (line 218) is CO₂ absorption in slag, isn't it? This is a minority process solution. Why don't you consider mainstream CCS, rather?
14. You assume that all CO₂ emissions can be captured into the slag. Is this right?
15. Ligne 266: why not quote the author, rather than the publisher. As far as I can remember, UNIDO does not take responsibility for the report.
16. Ligne 271: not clear if you distinguish between scrap/EAF and BF/OF routes at this level of your

discussion.

17. " BFS/Pig iron ratio and SS/Crude steel ratio will decrease at stable rates of 0.002/year and 0.001/year from 2020 to 2050". Source? Implicit assumptions?

18. Ligne 367: there the solution is clear, "we need to generate more slag". An oxymoron of course. But it stems from the formulation: " the emission reductions are mainly determined by the production of BFS and SS".

19. Ligne 374: replace thus by therefore

20. Ligne 389: the definition of the gasification process arrives late. What's it the TRL level?

21. Ligne 488: results and their detailed analysis is missing at this stage.

At the end of the paper, emissions are not really reduced by rather compensated. Is this equivalent, except from a bureaucratic standpoint?

Fig 20 and fol. Made to show scheme under various pathways in supplementary are very clear and could replace, mutatis, mutandis, some of the text in the main paper describing the pathways.

Reviewer #2 (Remarks to the Author):

The manuscript presents a study concerning potentials for reusing heat and recycling iron and steel CaO-rich waste to substitute emissions from the cement industry. It aims at finding an innovative solution for decarbonizing the steel sector from 2020-2050, by analysing the solution from technical and economic perspectives.

The manuscript is well written and organically developed. The state-of-the-art has been deeply analysed. The authors demonstrate a thorough knowledge of the topic developed in the manuscript. The adopted and applied methods are well and clearly described. The results are clearly presented also through very explanatory graphs.

However, in order to improve the quality of the manuscript, some minor revisions should be provided by the authors, that will make the manuscript suitable for publication:

- In the Abstract, please avoid references. The suggestion is to provide references in the manuscript, avoiding to group them (e.g. line 42, 56) but it is recommended to explain them in detailed and concise manner.

- Reference citations should be uniform. Please avoid different formats in the text, such as in lines 64, 118.

Reviewer #3 (Remarks to the Author):

Reviewer comments

Decarbonizing the iron and steel sector for a 2 °C target using inherent waste streams

Indeed, emissions from the iron and steel sector is huge compared to other industrial economic sectors, as such efforts towards the decarbonisation of the sector is of utmost importance. In this manuscript, the authors presented scenarios within a techno-economic analysis framework, exploring the inherent potential of waste streams based on high temperature slag when integrated with carbon capture and storage (CCS) mechanism. Under strong decarbonisation policy consistent with low warming targets, a CO₂ emission reduction of up to 28.5 ± 5.7% in steel and iron was estimated to be achievable by 2050 based on the energy recovery and resource recycling of glassy blast furnace slag and crystalline steel slag. A corresponding revenue of US\$35 and US\$40 billion globally in 2035 and 2050, respectively was also reported, which if invested in CCS could help meet emissions reduction targets by 2050. This is an important research to undertake and I therefore congratulate the authors. I, however, have the following suggestions/comments for the authors:

- Generally, the paper was difficult to read due to multiple referral to the supplementary information document, thereby affecting the overall flow, but I understand this pertains mainly to

restricted number of words.

- In line 56, the expression "CCS is the only option that would ultimately achieve a deep reduction in CO₂ emissions from this sector" is a bit of an overstatement. Different technology options offers different emission reduction potentials depending on the scenarios under consideration. The use of scrap-based EAFs with low carbon electricity has been touted as a viable option. High project costs, limited geological storage and public scrutiny has stalled development of CCS. These issues must be taken into consideration when assessing CCS whether as a standalone entity or as part of a wider systems. For CCS to be deployed at a commercial level, numerous issues including cost of implementation, monitoring and validation, regulations and legal aspects as well as public acceptance must resolved. Authors are encouraged to note these factors in their manuscript.
- Steel slags also finds a wide range of applications including agricultural fertilizers, road construction, soil improvements, etc. Authors are encouraged to briefly comment on this aspect and make a case as to why CCS is the preferred approach in comparison to other applications
- Lines 281 to 283, the authors stated, "Therefore, it is assumed that the annual BFS/Pig iron ratio and SS/Crude steel ratio will decrease at stable rates of 0.002/year and 0.001/year from 2020 to 2050, respectively". On what basis were these assumptions made?
- In terms of the cost benefit analysis (CBA) presented, authors listed all the parameters (e.g. capital, labour, energy maintenance etc) taken into consideration (expanded upon in the SI), yet there is no description or any mention of the actual type of CBA conducted and the equations linking all the parameters together. As such, it is difficult to ascertain how the CB figures were calculated. Were the calculations based on life cycle costing? Or on principles of marginal abatement cost curves (MACC)? Were the net present values (NPV) of the potential cost savings taken into consideration in the CBA? Overall, it is not clear at all the actual CBA framework adopted by the authors. Parameters for the CBA were listed and described but how they combine based on a defined calculation procedure is not stated. The calculation steps might be obvious to the authors in the supplementary figures, but may not be clear to the wider readership of Natur Comms. Essentially, an equation linking all the cost parameters together must be provided using an appropriate mathematical equation. For a paper that expresses cost gains as part of the key findings, the CBA presented is not rigorous enough.
- The style and language in which the Methods section was written should be revised. For instance, the authors stated: "To estimate the CO₂ emission reductions in different pathways in the timeframe of 2020-2050, the global outputs of crude steel should first be estimated". This and other statements should be revised.
- The calculation steps for the estimation of CO₂ emission reductions based on energy recovery from BFS and SS is basic. Authors are encouraged to adopt a more robust energy balance approach in this regard
- I am not sure about the requirements for supplementary document by Nature Comms, however, in the current submission, the supplementary document is 95 pages long. The first half of the paper is laced with linkages to the supplementary document, affecting flow and readability. The supplementary document should be as a supplement (as the name suggests) and not as the primary repository for the analysis and figures. If this is in line with the journal's policy, please ignore this comment
- In some portions of the paper, there are minor typos, which should be revised (e.g. raise instead of rise (line 4, pg. 2). Please correct all typos

Responses to the comments from Reviewer #1

General comment: *The paper addresses a relevant issue related to the GHG emissions of the steel sector and to mitigation solutions to achieve the Paris Agreement target (2°C).*

Our responses: We sincerely thank the reviewer for appreciating the relevance of our study. These comments and suggestions are very valuable for us to improve the study. We followed the recommendations and addressed each comment below.

Major comment 1: *It is very detailed and complex and, as such, is rather difficult to read by an expert in the field and probably confusing for readers from outside the field.*

The difficulty stems from the format of "article, appendices, supplementary materials comments, tables and figures", which are embedded into each other like Russian dolls and prevent a continuous reading of the paper's argument and claims. I understand that this fragmentation of the rhetoric argumentation of the paper is a requirement of the journal, not something that the authors particularly want to stick to.

Our responses: Because of the restricted length of manuscript, we put non-essential data and results as the supplementary information (SI) and only the main results are shown in text. In the revised manuscript, reference to SI has been reduced to keep clarity. To improve the readability of this paper, we have deleted some tables (Supplementary Tables 1-9 in the previous version) if the related data have been presented or partially presented in figures and some tables related to the calculation process (Supplementary Tables 22-26 in the previous version). Accordingly, all other tables are renumbered, and the related expressions have been revised.

Major comment 2: *The paper states in its conclusions that controlling the various kinds of slag produced in a steel mill in terms of energy and material recovery can provide enough revenue to pay for abatement technologies based on CCS and thus make it possible to achieve the 2° target. Not simply for a steel mill but for the whole steel sector, worldwide.*

This is a rather extraordinary claim. It exhibits a solution of which the steel sector is not aware and thus has a kind of magic character!

Now, let us analyze more in depth what we have understood of this complex paper and thus try

to express an opinion on the value of this claim.

The complexity of the paper stems from the fact that it examines a large set of technology-options for handling slag (BFS and SS), carries out cost analyses and does it for a time extension from 2020 to 2050 (foresight exercise).

The puzzling part of the paper is that focusing on a tiny part of steel production operations is enough to solve an otherwise difficult and forbearing problem (HGH mitigation).

Our responses: Heat recovery and resource recycling from high temperature slags represent a large part of potential for the emission reduction in the iron and steel sector, not a tiny part. 1) For the iron and steel industry, a substantial part of the energy used, 10-15% of the total input energy^{1,2}, remains in high temperature slag (1450-1650 °C), and almost none of this energy is recovered currently. 2) Further, most of the CaO for fluxing, by CaCO₃ calcination causing CO₂ emission, also remains in slag, and it is not well reused and recycled. Therefore, we show the importance to utilize waste heat and material resources in high temperature slag, as a potential to strongly cut carbon emission in the iron and steel industry. In this context, a treatment of high temperature of BFS and SS has a significant potential to contribute to the sectoral 2 °C target in the iron and steel industry. We analyze technological and economical feasibilities for the treatment of hot BFS and SS. **Firstly**, the sentences “*From the energy point of view, it is estimated that approximately 1700 MJ of heat is carried per tonne of BFS and SS.*” have been revised as “**From the energy point of view, energy carried by slag represents today 10-15% of the total energy input in the iron and steel industry^{15,16}.**” in **lines 66-67**.

Furthermore, in the revised manuscript, the uncertainties of the pathways and economic potentials are better discussed, as related to the sensitivity analyses. **(1)** the sentences “*....., considering the varying technological levels in the iron and steel industry and the uncertainties of global crude steel output^{3,5,6}.*” have been revised as “**....., considering uncertainties of technological levels in the iron and steel industry, the global crude steel output and related waste streams^{6,11-14}.**” in **lines 134-136**. **(2)** the sentences “*Along with the varying product prices, process costs and carbon price, the economics of a steel plant will change, and therefore, it is necessary to clarify the influences of these factors on a steel plant’s economics.*” have been revised as “**Economic uncertainty originates from the varying product**

prices, process and carbon price, estimated by sensitivity studies.” in lines 189-190. (3) the barriers and challenges that could prevent potentials to be fully reached are discussed based on the properties of BFS and SS, as detailed in the Discussion and conclusion section (more details referred to Responses to Major comments 3, 4 and 5). (4) to confirm the accuracy of the expressions, we revised the phrases like “achievement of 2 °C target emission reduction ratio” and “contributions to the 2 °C climate target” to a more objective style “CO₂ emission reduction ratio to the 2 °C target requirements” in the text, tables, and figures. Correspondingly, all the expressions related have been revised.

1. Barati, M., Esfahani, S. & Utigard, T. A. *Energy recovery from high temperature slags. Energy* 36(9), 5440–5449 (2011). (Numbered 15 in Revised Manuscript)

2. Zhang, H. et al. *A review of waste heat recovery technologies towards molten slag in steel industry. Appl. Energy* 112, 956–966 (2013). (Numbered 16 in Revised Manuscript)

Major comment 3: *Two technology pathways are examined: recovery of energy (heat) from hot slag and material recovery from the slag.*

Heat recovery has been a research and development theme in the steel sector, in Japan and Europe, for very many years and the search continues for practical solutions that can be implemented: many papers and grey literature reports were published but little operational feedback as the technologies are not yet mature enough for plant operative to invest in them and their economic benefits did not materialize. Therefore, the economic benefits that you show in the paper are difficult to match with what is actually going on in the real world. You probably overestimate the yield of energy recovery and its quality? And put too much expectation in the technologies? Thus the paper makes a long list of options (your pathways, which are fine), attaches a series of technologies to each one them and this produces a complex set of bullet points which describe options, some of them far-fetched, and most not necessarily feasible.

Regarding the recovery of material, mostly "calcium" for its hydraulic properties, we certainly agree. However, it seems to me that BF slag is 100% recovered and already sold to the cement industry in important parts of the world, in Europe for example. BOF slag is recovered for lesser applications, like roadmap and similar uses, which is paid less economically and

environmentally, but it is recovery and more ends up in slag heaps. EAF slag recovery is even less. The reason for the lower status of Steelmaking slag is its composition, which does not match the composition requirements of cement making: in this case, composition needs to be amended by using high-temperature processing which has been called slagmaking... a technique, which, anyway, remains confidential.

Our responses: The Reviewers makes good summary for the treatment of high temperature slag, which is mostly in consistent with our research. The main difference for the treatment of metallurgical slag including BFS and SS and other solid wastes is the high temperature state. Therefore, we need to extract the high-degree thermal heat from the hot slag, which is facing with great challenges currently. Our pathways and technology schemes are constructed strictly based on the fundamental properties of BFS and SS, including crystallization behaviours and glass forming abilities, while considering current technological levels and scenarios for future technology progress.

For the BFS treatment, three main approaches have been developed to cool hot BFS, namely natural cooling, water quenching, and dry granulation; the cooling rates of hot slag by these approaches are quite different, which determine the final states of the cooled BFS. For the current water-quenching method, cooled slag in a glassy state suitable for cement manufacturing can be obtained, while the thermal heat is totally wasted¹⁻³. Therefore, to obtain the glassy state using dry granulation method with effective heat recovery, is the most promising proposed technology, i.e., the integration of heat recovery and resource recycling. Technology advancement is important for realize all pathways in this study. Along with time going, some lab-scale technologies can be mature and industrialized; for example, using the heat in hot BFS to produce slag wool, has commercialized in China⁴ and dry granulation of BFS for the purpose of heat recovery and resource recycling is tried in the pilot scale in Australia⁵.

The SS treatment including BOF and EAF slag faces greater challenges because of its particular chemical compositions, as pointed by the Reviewer, like high basicity (CaO/SiO_2) and “FeO” concentration. SS does not quite match for cement making, at least directly used for that. We think that future treatment methods could be multiple while the main targets are to reuse the valuable compositions in the SS like CaO, “FeO” and P_2O_5 through effective

control of phase transformation and phase separation^{6,7}. The heat recovery from SS is another big challenge because of its high viscosity and crystallization behaviors, which makes it difficult to granulate into small droplets or particles. That is why in our study, cooled SS is assumed to be transformed in the crystalline state in both Pathways 5 and 6. This leads to a maximum potential that may be difficult to realize, as pointed out in the Discussion and conclusion Section in the revised manuscript.

To clarify the foregoing points related to BFS, **firstly**, the sentences “*Currently, three main approaches have been developed to cool hot BFS, namely, natural cooling, water quenching, and dry granulation; the cooling rates of these approaches are quite different, which determine the states of the cooled BFS.*” have been revised as “**Currently, three main approaches have been proposed to cool hot BFS with heat recovery and resource recycling, namely natural cooling, water quenching, and dry granulation, with quite different practical cooling rates and states of the cooled BFS.**” in **lines 98-100**. More details are discussed in Supplementary Note 1. **Secondly**, regarding the challenges for the treatment of SS, they are revised and discussed as “**The crystallization ability of SS is quite strong due to the high basicity (CaO/SiO₂) and “FeO” concentration, different from BFS. It is difficult to fully avoid crystallization behaviour of SS because of the high liquidus temperature. In this case, it will be challenging to obtain a 100% glassy state using a dry granulation method^{24,25}, as a crystalline state of SS is generally obtained.**” in **lines 104-107**. More details are discussed in Supplementary Note 2. **Thirdly**, in the Discussion and conclusion section, the barriers to treat BFS and SS and the related technological targets are further discussed in **lines 223-242**.

1. Barati, M., Esfahani, S. & Utigard, T. A. *Energy recovery from high temperature slags. Energy* 36(9), 5440–5449 (2011). (Numbered 15 in Revised Manuscript)
2. Zhang, H. et al. *A review of waste heat recovery technologies towards molten slag in steel industry. Appl. Energy* 112, 956–966 (2013). (Numbered 16 in Revised Manuscript)
3. Bisio, G. *Energy recovery from molten slag and exploitation of the recovered energy. Energy* 22, 501-509 (1997). (Numbered 17 in Revised Manuscript)
4. Cooksey, M., Guiraud, A., Kuan, B. & Pan, Y. *Design and operation of dry slag granulation pilot plant. J. Sustain. Metall.* 5, 181-194 (2019).
5. Zhao, D., Zhang, Z., Tang, X., Liu, L. & Wang, X. *Preparation of slag wool by integrated*

waste-heat recovery and resource recycling of molten blast furnace slags: from fundamental to industrial application. Energies 7, 3121-3135 (2014).

6. Engström, F., Adolfsson, D., Yang, Q., Samuelsson, C. & Björkman, B. *Crystallization behaviour of some steelmaking slags. Steel Res. Int. 81, 362-371 (2010). (Numbered 24 in Revised Manuscript)*

7. Yokoyama, K. et al. *Separation and recovery of phosphorus from steelmaking slags with the aid of a strong magnetic field. ISIJ Int. 47, 1541-1548 (2007). (Numbered 28 in Revised Manuscript)*

Major comment 4: *Therefore, at the end of the day, "Ca recovery" cannot be presented as a widely open option for the steel sector, as it is already in place in important regions of the world, where steel technology is quite advanced.*

Our responses: We thank the Reviewer for this important comment. The steel sector is trying to recover calcium from hot slag and the main issue is that the thermal energy in high temperature slag is totally wasted. In this study, we proposed two types of enhanced Ca recovery. The first type is the direct use of glassy BFS and part of SS as raw materials for cement manufacturing to replace the CaO and thus the CaCO₃ calcination, which has been realized in many countries¹⁻³. However, the optimization of slag cooling process is the main challenge to extract the thermal heat while obtaining a glassy state because currently the thermal energy in hot slag is wasted using water quenching method. Critical cooling rate is an important parameter to be modified to optimize the glass forming behaviours of the hot slag. The second type is the extraction of calcium-based materials from slag to be used as cycled agent for CCS⁴. The feasibility of enhanced Ca recovery depends on technology advancements in the mid to long-term. In this study, the former is mainly discussed. And economic analysis shows that obtaining glassy BFS accounts for the main target for BFS treatment using dry granulation method because only a glassy state of slag shows good hydraulic activity and economic value. The second type of Ca recovery is briefly discussed to give a material flow for the integration of CCS and treatment of waste streams.

To clarify the foregoing point, **firstly**, the sentences *"The achievements of the proposed pathways and schemes are greatly determined by the technological levels, including current*

levels and promising advancements in the mid- to long-term.” have been revised as “**The achievability of our proposed pathways and schemes are determined by technological improvements above current levels.**” in **lines 223-224**. **Secondly**, the sentences “*The integration of CCS and the re-use of waste streams can be realized from the perspective of material flow; i.e., after phase separation and post-treatment processes, CaO-based materials produced from BFS and SS can be used as raw materials for CCS (refs. ²⁵⁻²⁷).*” have been revised as “**The integration of CCS and the re-use of waste streams could be realized from material flow: after phase separation and post-treatment, the recovered calcium-based materials extracted from BFS and SS could be used for CCS²⁷.**” in **lines 215-217**.

1. Barati, M., Esfahani, S. & Utigard, T. A. *Energy recovery from high temperature slags. Energy* 36(9), 5440–5449 (2011). (Numbered 15 in Revised Manuscript)
2. Zhang, H. et al. *A review of waste heat recovery technologies towards molten slag in steel industry. Appl. Energy* 112, 956–966 (2013). (Numbered 16 in Revised Manuscript)
3. Bisio, G. *Energy recovery from molten slag and exploitation of the recovered energy. Energy* 22, 501-509 (1997). (Numbered 17 in Revised Manuscript)
4. Tian, S. et al. *Synthesis of highly efficient CaO-based, self-stabilizing CO₂ sorbents via structure-reforming of steel slag. Environ. Sci. Technol.* 49, 7464-7472 (2015). (Numbered 27 in Revised Manuscript)

Major comment 5: *Last but not least, selling BF slag as clinker additive to the cement sector did not generate enough revenue to make CCS possible! Extending the technique to more steel capacity in the world looks therefore more like technology transfer than the implementation of new breakthrough technology.*

Our responses: We thank the Reviewer for the constructive comment and deep thought. The use of BFS and SS in the cement manufacturing relies more on new technology breakthroughs for the treatment of them. For the treatment of BFS, the obtaining of glassy state using dry granulation method is the main R&D direction since currently, using the water quenching method, the thermal heat in the hot slag is totally wasted. For the treatment of SS, the control of phase transformation and phase separation, with effective heat recovery, is the most promising R&D direction. Therefore, new technologies to treat BFS and SS are quite needed

and still extensively exploited in the iron and steel sector; and we agree that technology transfer could be another issue for the implementation of new technologies. Secondly, the CCS scale we consider is determined by the net revenues generated by the treatments of BFS and SS, which greatly depends on the technological levels to treat BFS and SS in the iron and steel sector.

The technology breakthroughs needed for treating BFS and SS are detailed as “For the BFS treatment, the main technological challenge is to obtain a glassy state using a dry granulation method. This target is consistent with current technological developments for cooling hot BFS where extensive granulation processes have been developed¹⁵⁻¹⁷; for example, the concentrations of Al_2O_3 ²⁰ and CaO/SiO_2 ^{21,22} in BFS have been modified. For the SS treatment, recycling crystalline SS is the main challenge. The main valuable components include CaO, FeO and P_2O_5 , and their recycling relies on effective separation to enrich them in different phases. The main principles are to control the phase transformations during the cooling process by modifying the temperature schedule, atmosphere and material additions^{24,28}.” in the Discussion and conclusion section in **lines 224-232**.

Major comment 6: *The list of references is long and impressive, but centered either on east Asia papers (I cannot tell if there are many self-references or not) or on review papers from western papers. Moreover, there are clear gaps in the review of European papers and documents. And it is not clear what refers to operating practices in the steel sector and academic papers presenting models and schemes of technological solutions and principles, although the emphasis is on the latter rather than on the former one. Moreover you seem to be cherry-picking data among the various references.*

Our responses: For the references, we made sure of a non-biased survey of the literature. For the treatment of BFS and SS, more research especially in the lab are performed in Asia rather than European studies; the final reference list thus includes more references from Asia. This could be caused by that the output of crude steel and pig iron in Asian area is much larger than that in Europe. Based on the most recent database by the World Steel Association¹, in 2019, the output of crude steel in Asian area is 1,349 Mt, around seven times of that in European area, 196 Mt. Larger productions of crude steel and pig iron will produce more BFS and SS,

that has likely stimulated research on the treatment of high temperature BFS and SS in Asia.

For the parameters used for technological process and economic feasibility, as the Reviewer said, different sources are used, including: 1) results from pilot-scale trials such as for the dry granulation of BFS, the treatment of SS and those related to chemical gasification, and 2) market prices like the prices of glassy BFS and crystalline SS, electricity prices, etc. To improve this part of the study, **firstly**, we updated the reference list to include more references from Europe added as references 17, 23, 24 and 41. **Secondly**, the sentences “*First, because the prices of the input factors and products are quite different in various countries and areas, they are harmonized considering global income and labour levels³⁶⁻³⁸.*” have been revised as “**First, because the prices of input factors and products, obtained from the pilot-scale trials and market prices, are quite different in various regions, they are harmonized considering global income and labour levels³⁵⁻³⁷.**” in **lines 458-460**.

1. *World Steel Association, Statistical reports,*

<https://www.worldsteel.org/steel-by-topic/statistics/steel-statistical-yearbook.html>. (Numbered 14 in Revised Manuscript)

Specific comment 1. *After these general comments, more detailed questions are listed in what follows. “9% of annual global anthropogenic CO₂ emissions”: there is an emission range from 4% to 9%, depending on reference (anthropogenic emissions without or with ILUC) and level of specific emissions. Your references 1,2 are not very convincing!*

Our responses: As the Reviewer said, the CO₂ emission share from the iron and steel industry have a range of uncertainty. The current ratio was calculated based on the total CO₂ emission^{1,2} and the total anthropogenic emissions globally³. Totally, around 2866 Mt CO₂ is emitted in the iron and steel industry annually^{1,2} and meanwhile, the global CO₂ emission is around 36441 Mt per year³. Therefore, it can be calculated that the emission ratio of the iron and steel industry to the total global anthropogenic CO₂ emission is around 7.9%. To address the Reviewer’s comment, the sentences “*The decarbonization of the iron and steel industry, a sector contributing approximately 9% of annual global anthropogenic CO₂ emissions^{1,2} is challenged by persistently growing global steel demand and a lack of techno-economically feasible options for low-carbon steelmaking³⁻⁷.*” have been revised as “**The decarbonization of**

the iron and steel industry, contributing approximately 8% of current global anthropogenic CO₂ emissions is challenged by the persistently growing global steel demand and limitations of techno-economically feasible options for low-carbon steelmaking.” in lines 20-23. **Secondly**, the sentences “*The iron and steel industry is a particularly energy- and emission-intensive sector that accounts for ~9% of annual global anthropogenic CO₂ emissions, that is approximately 2800 Mt CO₂ per year^{1,2}.*” have been revised as “**The iron and steel industry is a particularly energy- and emission-intensive sector that accounts for about 8% of annual global anthropogenic CO₂ emissions, more than 2800 Mt CO₂ per year³⁻⁵.**” in lines 36-38. Furthermore, according to the Reviewer’s comment, references 1 and 2 have been deleted and replaced.

1. International Energy Agency. *Energy Technology Perspectives 2016: Towards Sustainable Urban Energy Systems*. <https://www.iea.org/topics/energy-technology-perspectives>. (Numbered 4 in Revised Manuscript)

2. Tian, S., Jiang, J., Zhang, Z. & Manovic, V. *Inherent potential of steelmaking to contribute to decarbonisation targets via industrial carbon capture and storage*. *Nat. Commun.* 9, 4422 (2018). (Numbered 5 in Revised Manuscript)

3. Global Carbon ATLAS, <http://www.globalcarbonatlas.org/fr/CO2-emissions> (Numbered 3 in Revised Manuscript)

Specific comment 2. “1.58 tons of CO₂ per ton of crude steel in 2020”: this is a low-value. Actual data (see the paper that you call UNIDO in your references) are much more dispersed and exhibit a higher average. It depends on the boundaries inside which emissions are accounted (e.g. are the coke ovens in or out? Etc.).

Our responses: We thank the Reviewer for this valuable comment. As the Reviewer pointed out, the average emission level per tonne of crude steel depends on the system boundaries and production processes. In consistent with the response to Specific comment 1, CO₂ emission intensity is calculated based on sectorial data given by the IEA^{1,2} (2866 Mt) and the production of crude steel and pig iron given by the UNIDO³ (1815Mt) (2866 /1815=1.58 tonnes of CO₂ per tonne of crude steel). In the revised manuscript, we account for the average of reported values. The sentences “*If current iron and steel production proceeds without the*

implementation of CO₂ emission reduction or Carbon Capture and Storage (CCS), the total emission budget in this sector by 2050 will exceed by a factor of two the limit proposed by the scenarios of refs.^{3,4}. Thus, substantial technological innovations are required to reduce the emissions intensities from 1.58 tonnes of CO₂ per tonne of crude steel in 2020 to 0.52 tonnes of CO₂ per tonne of crude steel in 2050 so that the two emission scenarios^{3,4} can be consistent with future production projections^{4,5} (Fig. 1). ” have been revised as “If the current iron and steel production proceeds without the implementation of CO₂ emission reduction or carbon capture and storage (CCS), the total emission budget in this sector by 2050 will exceed by a factor of two the limit proposed by refs^{4,5}, equivalent to a reduction of the emission intensity from 1.58 tonnes of CO₂ per tonne of crude steel in 2020 down to a target 0.52 tonnes of CO₂ per tonne in 2050 (Fig. 1). ” in lines 42-47.

1. International Energy Agency. *Energy Technology Perspectives 2016: Towards Sustainable Urban Energy Systems*. <https://www.iea.org/topics/energy-technology-perspectives>. (Numbered 4 in Revised Manuscript)

2. Tian, S., Jiang, J., Zhang, Z. & Manovic, V. *Inherent potential of steelmaking to contribute to decarbonisation targets via industrial carbon capture and storage*. *Nat. Commun.* 9, 4422 (2018). (Numbered 5 in Revised Manuscript)

3. Birat, JP., ArcelorMittal Global R & Maizières-lès-Metz, D., UNIDO. *Steel Sectoral Report: Contribution to the UNIDO Roadmap on CCS –Fifth Draft*. (Numbered 6 in Revised Manuscript)

Specific comment 3. *The list of low-carbon new technology listed lines 54 and fol. Is a bit strange: TRT are very common in Asia, where electricity is expensive, start to appear in Europe; CCPPs are not relevant here (by the way, are power plants inside "the boundaries" or not); and COREX definitely does not reduce GHG emissions, quite the contrary, and, anyway, there are so few of them in the world.*

Our responses: According to the Reviewer’s comment, the techniques including TRTs, CCPPs and Corex are at different developing stages; for example, TRTs have been realized in many iron and steel plants in Asia. Here we list these techniques just to give a broad technique portfolio for the iron and steel industry. We acknowledge that Corex does not reduce GHG

emissions, as pointed by the Reviewer; therefore, it has been deleted in this study. The gradual commercialization of these techniques will need time but could also contribute to the achievement of the sectoral 2 °C target, in addition to the treatment of waste streams, BFS and SS discussed in this study. In order to clarify this point, **firstly**, sentences “*Despite the research and development in emerging CO₂ abatement technologies in the iron and steel sector, including top gas pressure recovery turbines (TRTs), combined cycle power plants (CCPPs) and Corex processes¹⁰*” have been revised as “**Some production technologies began to be implemented to reduce CO₂ emission, like top gas pressure recovery turbines (TRTs) and combined cycle power plants (CCPPs)⁹.**” in **lines 52-54**. **Secondly**, the sentences “*This target requires the coupling of CCS to the iron and steel production, and technological innovations including TRTs, CCPPs, Corex to finally reduce*” have been revised as “**This target requires the coupling of CCS to the iron and steel production in different steps such as in mainstreams, and new low carbon production technologies like TRTs and CCPPs to finally reduce the emission intensity to the target of 0.52 tonnes of CO₂ per tonne of crude steel in 2050⁴⁻⁶.**” in **lines 206-209**.

Specific comment 4. *Line 65: large internal potential is not quite true, at least where advanced world class steel mills are concerned; "valorized" is an open question, as the reason why so many technologies have been experimented upon is that they did not provide enough revenue (ROI) to be implemented industrially.*

Our responses: Based on both our previous and current research experiences and the reports cited, the treatment of BFS and SS is a key issue for the steel mills globally, regarding energy recovery and resource recycling. This greatly determines the low carbon pathways for the iron and steel industry because slag naturally accumulated in the slag heaps causes not only energy and resource waste but also environmental pollutions. The treatment of BFS and SS covers a series of technologies like granulation, chemical gasification and phase separations, which have quite different technological readiness states. For example, water quenching of BFS have been widely used, and dry granulation of BFS has been tested in the pilot scale; while chemical gasification is mainly investigated in the lab. Therefore, the realization of the pathways proposed this study depends on the cost and scalability of these technologies in the

near-, mid- and long-term future. To clarify the foregoing points, the sentences “*Accordingly, five schemes are proposed here to engineer Pathways 5 and 6, physical granulation*” have been revised as “**Accordingly, five schemes with different TRLs are proposed here to engineer Pathways 5 and 6, namely physical granulation**” in **lines 149-150**.

Specific comment 5. *Line 71: this is already practiced and for a very long time (> 50 years?). On the other hand, to which industry should the reduction of emissions be allocated? Steel or cement? The issue is unresolved, as the cement industry is not willing to yield: compare the price of cement to the price of CO₂!*

Our responses: We thank the Reviewer for this valuable comment. Through the utilization of cooled slag especially the glassy BFS as the raw materials in the cement manufacturing, the iron and steel industry and the cement industry are integrated, namely the material flow. In our study, we assumed that the emission reduction by the treatment of BFS and SS is allocated in the iron and steel industry. In the future research, we can further investigate the emission reduction potential in integrated iron and steel with cement industry connected by Ca material flow. For the cement manufacturing, around 0.9 tonnes of CO₂ is emitted per tonne of cement¹ and here the same CO₂ price is used, namely 30 US\$/tonne. The cement prices are quite different estimated from different models²⁻⁴; for example, it has been reported⁴ in 2019 the average price of gray cement is around US\$60/tonne of cement. Using the emission intensity, we can calculate the ratio of CO₂ price to the cement price is $30 \times 0.9 / 60 = 0.45$. This could clearly prove that the CO₂ price could account for one of the main costs for cement industry.

To clarify the foregoing points, **firstly**, the sentences “*If BFS is in the glassy state, it can be used for cement manufacturing to replace limestone calcination; however, if it is in the crystalline state, the economic value is significantly reduced¹³⁻¹⁵.*” have been revised as “**If BFS is in the glassy state, it can be used as a cement feedstock and substituted to limestone calcination CO₂ emissions, which has a mean emission intensity of ~0.9 tonnes of CO₂ per tonne of cement²³.**” in **lines 89-92**. **Secondly**, the sentences “**Here we assumed that CO₂ emissions avoided in the cement production can be claimed by the steel industry, although other accounting possibilities are open because cement and steel produces will need to be cooperate and integrate their activities to make this reduction happen.**” have been added in

lines 93-96.

1. Naqi, A. & Jang, J.G. *Recent progress in green cement technology utilizing low-carbon emission fuels and raw materials: A review. Sustainability 11, 537 (2019). (Numbered 23 in Revised Manuscript)*
2. Ilbeigi, M., Ashuri, B. & Joukar, A. *Time-series analysis for forecasting asphalt-cement price. Journal of Management in Engineering 33, 04016030 (2017).*
3. Kamar, K. *Analysis of the effect of return on equity (ROE) and debt to equity ratio (DER) on stock price on cement industry listed in Indonesia stock exchange (IDX) in the year of 2011-2015. IOSR Journal of Business and Management 19, 66-76 (2017).*
4. *Cemweek, WORLD CEMENT PRICES CONTINUE TO SLIDE IN 1Q2019. See also: <https://www.cwgrp.com/cemweek-features/516086-world-cement-prices-continue-to-slide-in-1q2019>.*

Specific comment 6. *Line 74: where do these estimates (potential emission reduction) come from? As a matter of fact, I did not understand how you extrapolated costs to 2030 and 2050. For example, you assume the cost of CO₂ to be 30€/t today. How high will they be in 2050? They'll remain at 30 or rise to 500, as some literature work suggest?*

Our responses: For the process analysis where the main products are calculated, it is mainly based on energy balance and mass balance. For the economic analysis, the parameters used are mainly based on the references with reasonable assumptions. Here constant values of these parameters are employed for calculations. Considering the possible change of the different parameters, a sensitivity analysis is conducted where $\pm 50\%$ change is used.

Regarding the cost of CO₂ pointed by the Reviewer, it is in a wide range¹⁻⁵ and here a constant value of US\$30 per tonne of CO₂ is used as the reference level. We consider two types of costs related to CO₂ emission or reduction in this study, namely carbon price due to CO₂ emission and CO₂ avoided cost/benefit by CCS. In this study, we select the same value for these two costs to simplify the analysis, i.e., US\$30 per tonne of CO₂. As pointed by the Reviewer, this value could change with the technology advancements and the construction of carbon trading market, i.e., the carbon price due to CO₂ emission could vary a lot^{1,2} while the CO₂ avoided cost by CCS could decrease³⁻⁵. Therefore, a sensitivity analysis is performed where $\pm 50\%$ change is used (Fig. 2d), and we found the carbon price has limited effect on the

economics of a steel plant in the current analyses based on the treatment of waste streams.

To clarify the foregoing points, **firstly**, the sentences “*Along with the varying product prices, process costs and carbon price, the economics of a steel plant will change, and therefore, it is necessary to clarify the influences of these factors on a steel plant’s economics.*” have been revised as “**Economic uncertainty originates from the varying product prices, process and carbon price, estimated by sensitivity studies.**”, in **lines 189-190**. **Secondly**, the sentences “*Regarding the economics of a steel plant, for the various schemes, the market situation will change with the varying operational conditions over time such as the material prices and labour costs.*” have been revised as “**Regarding the economics of a plant, for the various schemes, the market situation will change with the varying operational conditions in the mid-to long-term such as the material prices, labour cost and CO₂ price.**” in **lines 477-479**. **Thirdly**, to keep the accuracy of the expressions in our study, we have revised the words “CO₂ tax” to “CO₂ price” and correspondingly, the related cost and benefits have been revised as “CO₂ cost” and “CO₂ avoided benefits” in the text, figures, and tables.

1. Bauer, N., et al., *Quantification of an efficiency–sovereignty trade-off in climate policy*. *Nature* 588(7837), 261-266 (2020). (Numbered 41 in Revised Manuscript)

2. Vrontisi, Z. et al. *Enhancing global climate policy ambition towards a 1.5 °C stabilization: a short-term multi-model assessment*. *Environ. Res. Lett.* 13, 044039 (2018).

3. Tian, S., Jiang, J., Zhang, Z. & Manovic, V. *Inherent potential of steelmaking to contribute to decarbonisation targets via industrial carbon capture and storage*. *Nat. Commun.* 9, 4422 (2018). (Numbered 5 in Revised Manuscript)

4. Birat, JP., ArcelorMittal Global R & Maizières-lès-Metz. D., UNIDO. *Steel Sectoral Report: Contribution to the UNIDO Roadmap on CCS –Fifth Draft*. (Numbered 6 in Revised Manuscript)
<https://citeseerx.ist.psu.edu/viewdoc/download?doi=10.1.1.466.1352&rep=rep1&type=pdf>.

5. Budinis, S. et al. *An assessment of CCS costs, barriers and potential*. *Energy Strateg. Rev.* 22, 61-81 (2018). (Numbered 26 in Revised Manuscript)

Specific comment 7. Line 86: *it sounds like a good idea to invest the revenue (if there is any!) of low-carb technologies to carbon mitigation. However, this is not necessarily how a steel company operates. Usually revenues are not tagged to any specific expenditures.*

Our responses: In this study, we conduct the cost-benefit analysis of a steel plant during the treatment of BFS and SS, and the net revenues of this plant with an annual crude steel output of 1 Mt are obtained. An assumption is made that these net revenues are used for CCS available at an average cost, i.e., under a strong climate policy (carbon price) constraint. To keep the accuracy of the expressions, the sentences “*We find that using the iron and steel reduction technological potentials and cost analysis quantified in this study, US\$35 billion and US\$40 billion revenues could be generated globally in 2035 and 2050 (Method M4). Re-investing those financial benefits into CCS integrated into manufacturing processes could achieve deep decarbonization of the iron and steel sector before 2050 consistent with the 2 °C emission scenario³⁻⁵, without any external investments (for more details, refer to Methods and Supplementary Notes).*” have been revised as “**We find from this analysis that net revenues of US\$35 billion and US\$40 billion could be generated globally in 2035 and 2050. A re-investment of these revenues into CCS coupled to manufacturing processes could reduce equivalent CO₂ emissions before 2050 to be consistent with the sectoral 2 °C target of refs^{4,5} (Methods and Supplementary Notes). Meeting this condition will require a very strong policy to decarbonize the sector, with incentives and regulation to re-invest into CCS.**” in **lines 81-86**.

Specific comment 8. *Line 107: long-time held \diamond held for a long time?*

Our responses: The expressions have been revised from “*long-time held*” to “**held for a long time**” in **line 109**.

Specific comment 9. *Line 130: a consequential approach is missing in the argument. If slag is added to clinker, then it may replace Portland cement but also add to overall cement production and thus not substitute anything, cf. the well-known Braess's paradox in road traffic modeling, whereby when more lanes are added to a freeway, traffic increases rather than became smoother.*

Our responses: For the BFS treatment, one of the main utilizations of glassy BFS is to use it as raw materials for cement manufacturing to replace the lime and limestone calcination, which will reduce the carbon emission per tonne cement produced; while the use of the

calcium resource in SS relies on the further technology innovations especially those for phase transformations and separations. Because the cement output globally (over 4 billion tonnes¹) is much larger than the discharge of BFS and SS, the utilization of the slag for cement manufacturing will therefore not add the overall production of cement but mainly depending on the technology advancements regarding the treatment of BFS and SS. To clarify the foregoing points, the sentences “Slags in a glassy state, with high hydraulic activity, can be used as raw materials for cement manufacturing, which accounts for the dominant resource recycling of BFS and SS.” have been revised as “Slag in a glassy state, with high hydraulic activity, can be used as raw materials for cement manufacturing considering the huge consumption of cement accompanied with the expanding urbanization and industrialization globally, which accounts for the dominant resource recycling of BFS and SS¹⁵⁻¹⁷.” in **lines 317-321**.

I. Naqi, A. & Jang, J.G. Recent progress in green cement technology utilizing low-carbon emission fuels and raw materials: A review. Sustainability 11, 537 (2019). (Numbered 23 in Revised Manuscript)

Specific comment 10. *As pointed out before, blast furnace and steelmaking slags are quite different.*

Our responses: We agree that blast furnace slag and steelmaking slag are produced in different processes, and their chemical compositions and properties like crystallization behaviours and glass forming abilities are quite different. Therefore, the treatment of these two types of slag should be very different, which are detailed in the **Section of “Understanding the waste streams properties to re-use them”** and **Supplementary Note 1** and **Note 2**. To further highlight this difference, the sentences “For SS, its crystallization ability is quite strong due to the high basicity and “FeO” concentration. Therefore, it is difficult to fully avoid crystallization behaviour because of the high liquidus temperature. In this case, it is challenging to obtain a 100% glassy state using a dry granulation method²¹⁻²³, and a crystalline state of SS is generally obtained.” have been revised as “The crystallization ability of SS is quite strong due to the high basicity (CaO/SiO₂) and “FeO” concentration, different from BFS. It is difficult to fully avoid crystallization behaviour of SS because of the

high liquidus temperature. In this case, it will be challenging to obtain a 100% glassy state using a dry granulation method^{24,25}, as a crystalline state of SS is generally obtained.” in lines 104-107.

Specific comment 11. *Line 196: "The labor, maintenance and energy costs all have limited influences on the economics of a steel plant": this is what your graphs shows but it is very counterintuitive. On the other hand, the results are very sensitive to the price of by-products, which is also counter intuitive.*

Our responses: The details of economic analysis are shown in Supplementary Tables 6-10 and the economic sensitivity analyses are shown in Fig. 2d and Supplementary Figs. 15-18 for technological Schemes 1-5. Because of the large output coupled with optimal treatment of BFS and SS in a steel plant, the average labour, maintenance and energy costs (per tonne) end up to be relatively low compared to the price of re-useable glassy BFS and crystalline SS. Therefore, the effects of these factors are smaller than the price of the by-products. This could be a significant advantage for a future steel plant as the necessary technologies are fully developed. To clarify the foregoing points more clearly, the sentences “*The labour, maintenance and energy costs all have limited influences on the economics of a steel plant.*” have been revised as “**Comparatively, the effects of labour, maintenance and energy costs are smaller than the revenues from better re-use of slag products in our analysis. We acknowledge uncertainties in estimating labour and maintenance for slag recovery technologies, as these solutions are not implemented at scale today.**” in lines 195-199.

Specific comment 12. *Line 210, Corex et les autres procédés sont déjà cités.*

Our responses: As detailed in the response to Specific comment 3, Corex, as pointed by the Reviewer, has been discarded in this study for its more intensive GHG intensity.

Specific comment 13. *CCS (line 218) is CO₂ absorption in slag, isn't it? This is a minority process solution. Why don't you consider mainstream CCS, rather?*

Our responses: Because slag is CaO-rich materials, if the CaO could be extracted through advanced methods, it can be used for carbon capture and reused many times¹. The CCS can be

also introduced to the mainstream of ironmaking and steelmaking process, as pointed out by the Reviewer, where investments could be partially supplied by the treatment of waste streams. To clarify the foregoing points, **firstly**, the sentences “*This target requires the coupling of CCS to the iron and steel production, and technological innovations including TRTs, CCPPs, Corex to finally reduce*” have been revised as “**This target requires the coupling of CCS to the iron and steel production in different steps such as in mainstreams, and new low carbon production technologies like TRTs and CCPPs to finally reduce the emission intensity to the target of 0.52 tonnes of CO₂ per tonne of crude steel in 2050⁴⁻⁶.**” in **lines 206-209**. **Secondly**, the sentences “*The integration of CCS and the re-use of waste streams can be realized from the perspective of material flow; i.e., after phase separation and post-treatment processes, CaO-based materials produced from BFS and SS can be used as raw materials for CCS (refs. 25-27).*” have been revised as “**The integration of CCS and the re-use of waste streams could be realized from material flow: after phase separation and post-treatment, the recovered calcium-based materials extracted from BFS and SS could be used for CCS²⁷.**” in **lines 215-217**.

1. Tian, S. et al. *Synthesis of highly efficient CaO-based, self-stabilizing CO₂ sorbents via structure-reforming of steel slag. Environ. Sci. Technol.* 49, 7464-7472 (2015). (Numbered 27 in Revised Manuscript)

Specific comment 14. *You assume that all CO₂ emissions can be captured into the slag. Is this right?*

Our responses: We do not assume that all CO₂ emissions can be captured into the slag. As detailed in the Responses to Major comment 4 and Specific comment 13, for the utilization of calcium-based materials extracted from hot slag, CO₂ emissions are not directly captured into the slag. Instead, a possible flow of the integration of slag treatment with CCS is that the CaO is extracted from the slag and the prepared calcium-based materials used as cycled agent for CCS. Another method is to use the slag to directly capture and mineralize the CO₂ where the steelmaking slag can be used only for one time, as reported in **references 26 and 27**. This strategy is not discussed in detail in this study. To avoid the possible misunderstanding, **references 26 and 27** have been deleted from the reference list since they directly used slag to

mineralize the CO₂ but not use slag to extract calcium-based materials for CCS. Accordingly, all other references in the reference list have been renumbered.

26. Mo L. Carbon dioxide sequestration on steel slag, *Carbon Dioxide Sequestration in Cementitious Construction Materials*, Jan 1, 2018, 175-197, Woodhead Publishing.

27. Pan, S. Y. et al. CO₂ mineralization and utilization using steel slag for establishing a waste-to-resource supply chain. *Sci. Rep.* 7, 1-11 (2017).

Specific comment 15. Line 266: *why not quote the author, rather than the publisher. As far as I can remember, UNIDO does not take responsibility for the report.*

Our responses: In the reference list, we have revised the quote to “6. Birat, JP., ArcelorMittal Global R & Maizières-lès-Metz. D., UNIDO. Steel Sectoral Report: Contribution to the UNIDO Roadmap on CCS –Fifth Draft. <https://citeseerx.ist.psu.edu/viewdoc/download?doi=10.1.1.466.1352&rep=rep1&type=pdf>” in lines 507-509.

Specific comment 16. Line 271: *not clear if you distinguish between scrap/EAF and BF/OF routes at this level of your discussion.*

Our responses: Scrap/EAF and BF/BOF routes are two main processes for the steel industry, which is also indicated by the ratio of Pig iron to Crude steel. In order to clarify the foregoing point, **firstly**, the sentences “*Based on the outputs of crude steel and pig iron between 2008 and 2018, this parameter is in the range of 0.71~0.75 (ref. ³¹; Supplementary Table 12).*” have been revised as “. **Based on the outputs of crude steel and pig iron between 2008 and 2018, this parameter is in the range of 0.71~0.75 considering the two main processes of scrap-based electric arc furnace (scrap/EAF) and blast furnace/basic oxygen furnace (BF/BOF) in the iron and steel industry^{9,11,14} (Supplementary Table 3).**” in lines 261-264. **Secondly**, the sentences “*Due to technological advancements and the utilization of scrap in steelmaking, this ratio can be slightly decreased¹⁰.*” have been revised as “**Due to technological progress and the increasing utilization of scrap/EAF process, this ratio could be slightly decreased in the future^{9,11,14}.**” in lines 264-266.

Specific comment 17. *"BFS/Pig iron ratio and SS/Crude steel ratio will decrease at stable rates of 0.002/year and 0.001/year from 2020 to 2050". Source? Implicit assumptions?*

Our responses: The discharge ratios of BFS/Pig iron and SS/Crude steel are assumed based on data collected in the past years. There are very limited research papers accurately reporting the discharge outputs of BFS and SS or predicting the future BFS and SS productions especially in the mid- to long-term.

For example, regarding the BFS/Pig iron ratio in the past years, we can use the outputs of pig iron collected by the World Steel Association¹ and discharges of BFS summarized from the Global Slag Review^{2,3}. In 2018, it was reported 333 Mt BFS was discharged² with the pig iron output of 1253 Mt¹; it can be calculated that the BFS/Pig iron in 2018 was around 0.266. In 2017, it was reported 360 Mt BFS was discharged² with a pig iron output of 1186 Mt¹, which gives a BFS/Pig iron ratio of 0.304. Based on these estimations, we can see the great decrease of BFS/Pig iron ratio. The discharges of BFS and SS were also collected from the U.S. Geological Survey⁴. However, a large range is provided; for example, 320~384 Mt BFS and 190-280 Mt SS were discharged in 2019; accordingly, the ranges of BFS/Pig iron and SS/Crude steel were 0.25~0.30 and 0.10~0.15, respectively. In this study, a decreasing and stable BFS/Pig iron ratio was used to simplify the analysis considering possible technological progress to reduce this ratio in the future (similarly to SS/Crude steel ratio).

To clarify the foregoing analysis, **firstly**, the sentences *"In the current manufacturing process of the iron and steel sector, producing one tonne of crude steel generates considerable waste streams, including 250-350 kilograms of blast furnace slag (BFS) at temperatures of 1500-1600 °C and 100-200 kilograms of steel slag (SS) at temperatures of 1550-1650 °C (ref. ¹²)." have been revised as "In the current manufacturing process of iron and steel, producing one tonne of crude steel generates waste streams in the range of 250-300 kilograms of blast furnace slag (BFS) at temperatures of 1500-1600 °C and 100-150 kilograms of steel slag (SS) at temperatures of 1550-1650 °C¹¹⁻¹⁴."* in **lines 61-64**. **Secondly**, the sentences *"At the current global level in 2020, these two parameters are assumed to be approximately 0.260 and 0.130, respectively¹²⁻¹⁵. Due to technological advancements, the discharge of BFS and SS will decrease continuously¹³⁻¹⁵. Therefore, it is assumed that the annual BFS/Pig iron ratio and SS/Crude steel ratio will decrease at stable rates of 0.002/year and 0.001/year from 2020 to*

2050, respectively.” have been revised as “At the current global level in 2020, these two parameters are approximately 0.26 and 0.13, respectively¹¹⁻¹⁴. Due to technological progress, the discharge of BFS and SS could decrease continuously^{12,13}. Therefore, it is assumed that the BFS/Pig iron and SS/Crude steel ratios will decrease at rates of 0.002/year and 0.001/year from 2020 to 2050, respectively¹¹⁻¹⁴.” in lines 272-276. Moreover, the references have been updated correspondingly.

1. World Steel Association, *Statistical reports*,

<https://www.worldsteel.org/steel-by-topic/statistics/steel-statistical-yearbook.html> (Numbered 14 in Revised Manuscript)

2. Global Slag, *Global Slag 2018 Review*:

<https://www.globalslag.com/conferences/global-slag/review/global-slag-review-2018>
(Numbered 12 in Revised Manuscript)

3. Global Slag, *Global Slag 2017 Review*:

<https://www.globalslag.com/conferences/global-slag/review/global-slag-review-2017>
(Numbered 13 in Revised Manuscript)

4. U.S. Geological Survey,

https://www.usgs.gov/centers/nmic/iron-and-steel-slag-statistics-and-information?qt-science_support_page_related_con=0#qt-science_support_page_related_con (Numbered 11 in Revised Manuscript)

Specific comment 18. Line 367: there the solution is clear, "we need to generate more slag". An oxymoron of course. But it stems from the formulation: "the emission reductions are mainly determined by the production of BFS and SS".

Our responses: The potential of emission reduction is indeed influenced by several variables, including the production of BFS and SS and their heat capacities and discharge temperatures. This is detailed by the sensitivity analysis in this study. Based on the current and future technology levels, there will still be enough slag despite their decreasing ratio to pig iron and steel, as detailed in Response to Specific Comment 17, to make significant CO₂ reductions with re-use technologies. The sentences “..... it can be observed that the emission reductions are mainly determined by the production of BFS and SS, and their heat capacities and

discharge temperatures;” have been revised as “..... it can be observed that the emission reductions are influenced by the production of BFS and SS and their heat capacities and discharge temperatures;” in **lines 360-362**.

Specific comment 19. *Line 374: replace thus by therefore.*

Our responses: The expression has been revised in **line 367**. Furthermore, other similar expression like that in **line 266** have also been revised.

Specific comment 20. *Line 389: the definition of the gasification process arrives late. What's it the TRL level?*

Our responses: For the physical method, part of it like water quenching of BFS has been realized commercially, and chemical methods are only under development in the lab-scale¹⁻³. To express the gasification process earlier in the text, **firstly**, the sentences “*while for the chemical gasification method, the selection of granulation and gasification agents is the key issue.*” have been revised as “**while chemical gasification methods are only developed as laboratory research with a low technologies readiness level (TRL)¹⁵⁻¹⁷, the selection of granulation and gasification agents is the key issue.**” in **lines 147-149**. **Secondly**, to clarify the TRL level, the sentences “*For the gasification method, three types of agents can be used, namely.....*” have been revised as “**For the gasification methods currently of low TRL¹⁵⁻¹⁷, three types of agents can be used, namely**” in **lines 384-385**.

1. Barati, M., Esfahani, S. & Utigard, T. A. *Energy recovery from high temperature slags.*

Energy 36(9), 5440–5449 (2011). (Numbered 15 in Revised Manuscript)

2. Zhang, H. et al. *A review of waste heat recovery technologies towards molten slag in steel industry.* *Appl. Energy 112, 956–966 (2013). (Numbered 16 in Revised Manuscript)*

3. Bisio, G. *Energy recovery from molten slag and exploitation of the recovered energy.*

Energy 22, 501-509 (1997). (Numbered 17 in Revised Manuscript)

Specific comment 21. *Line 488: results and their detailed analysis is missing at this stage.*

Our responses: The analysis of economics and economic sensitivity is detailed in the **Section of “Economics of the steel sector”** and not in the Method Section. To effectively relate these

different sections, the sentences “Accordingly, the economic sensitivity at the plant level is finally analysed for an annual crude steel output of 1 Mt.” have been revised as “Accordingly, the economic sensitivity at the plant level is finally analyzed for an annual crude steel output of 1 Mt (detailed in the section of “Economics of the steel sector”).” in lines 489-490.

Specific comment 22. *At the end of the paper, emissions are not really reduced by rather compensated. Is this equivalent, except from a bureaucratic standpoint?*

Our responses: The emissions are reduced in the iron and steel industry based on the heat recovery and resource recycling of hot slag. Firstly, the results in this study are firstly discussed based on the solid analyses of the global technologies developed for the treatment of waste streams, high temperature BFS and SS, in the iron and steel industry, without any bureaucratic standpoint. Secondly, an integrated emission reduction strategy is used: the thermal heat in the hot slag is recovered, which can be reused as energy in the iron and steel sector to reduce that produced by fossil fuel, and the material resources especially CaO in the slag is used to replace the limestone calcination and therefore reduce the CO₂ emission.

Specific comment 23. *Fig 20 and fol. Made to show scheme under various pathways in supplementary are very clear and could replace, mutatis, mutandis, some of the text in the main paper describing the pathways.*

Our responses: We thank the Reviewer for this good suggestion. To highlight the relationship between engineering scheme and pathway, a new figure has been added into the Supplementary information, shown in the right side. The new figure is numbered **Supplementary Fig. 20**. Accordingly, the related expressions have been added in **line 153** and **line 644**, and all the other figures in the **Supplementary information** have been renumbered.

Pathway 5 + Pathway 6	
Scheme 1	Physical Granulation
Scheme 2	Air granulation + CO ₂ gasification
Scheme 3	Air granulation + H ₂ O gasification
Scheme 4	CO ₂ granulation + CO ₂ gasification
Scheme 5	CO ₂ granulation + CO ₂ /H ₂ O gasification

The key information of these technological schemes is also provided in the main text. **First**, to briefly express the Scheme 1, the sentences “*For Scheme 1, three steps make up the whole process.....*” have been revised as “**For Scheme 1, three steps make up the whole process including air-slag granulation, air-slag heat transfer and air-steam heat transfer**” in **lines 420-421**. **Second**, to briefly introduce Schemes 2-5, the sentences “*For Schemes 2-5, the energy balance of the whole process is expressed by the following equation.....*” have been revised as “**For Schemes 2-5, the energy balance of the whole process, mainly composed of the granulation and gasification steps, is expressed by the following equation**” in **lines 438-439**.

Responses to the comments from Reviewer #2

General comments:

The manuscript presents a study concerning potentials for reusing heat and recycling iron and steel CaO-rich waste to substitute emissions from the cement industry. It aims at finding an innovative solution for decarbonizing the steel sector from 2020-2050, by analysing the solution from technical and economic perspectives.

The manuscript is well written and organically developed. The state-of-the-art has been deeply analysed. The authors demonstrate a thorough knowledge of the topic developed in the manuscript. The adopted and applied methods are well and clearly described. The results are clearly presented also through very explanatory graphs.

However, in order to improve the quality of the manuscript, some minor revisions should be provided by the authors, that will make the manuscript suitable for publication.

Our responses: We sincerely thank the Reviewer for the positive comments and the constructive suggestions. The responses to the specific comments and the revisions are listed as follows point-by-point.

Specific comment 1. *In the Abstract, please avoid references. The suggestion is to provide references in the manuscript, avoiding to group them (e.g. line 42, 56) but it is recommended to explain them in detailed and concise manner.*

Our responses: We deleted the references in the Abstract and accordingly, the references in the text have been renumbered. Moreover, according to the Reviewer's suggestion, we have slightly adjusted the references and revised the expressions related to the literature, if necessary, to decrease their grouping. **Firstly**, the sentences related to the treatment of BFS have been revised from “*This target agrees with current technological developments for cooling hot BFS where extensive granulation processes have been developed¹³⁻¹⁶.*” to “**This target is consistent with current technological developments for cooling hot BFS where extensive granulation processes have been developed¹⁵⁻¹⁷; for example, the concentrations of Al_2O_3 ²⁰ and CaO/SiO_2 ^{21,22} in BFS have been modified.**” in **lines 225-228**. **Secondly**, regarding the treatment of SS, the sentences “**Recently, other applications of SS such as agricultural fertilizers²⁹ and soil improvement agents³⁰ have also been developed.**” have been added in

lines 232-233.

Specific comment 2. *Reference citations should be uniform. Please avoid different formats in the text, such as in lines 64, 118.*

Our responses: The formats of the citations were all harmonized.

Responses to the comments from Reviewer #3

General comments:

Indeed, emissions from the iron and steel sector is huge compared to other industrial economic sectors, as such efforts towards the decarbonisation of the sector is of utmost importance. In this manuscript, the authors presented scenarios within a techno-economic analysis framework, exploring the inherent potential of waste streams based on high temperature slag when integrated with carbon capture and storage (CCS) mechanism. Under strong decarbonisation policy consistent with low warming targets, a CO₂ emission reduction of up to 28.5 ± 5.7% in steel and iron was estimated to be achievable by 2050 based on the energy recovery and resource recycling of glassy blast furnace slag and crystalline steel slag. A corresponding revenue of US\$35 and US\$40 billion globally in 2035 and 2050, respectively was also reported, which if invested in CCS could help meet emissions reduction targets by 2050. This is an important research to undertake and I therefore congratulate the authors. I, however, have the following suggestions/comments for the authors.

Our responses: We sincerely thank the Reviewer for the positive comments and the valuable and constructive suggestions. The responses to the specific comments and the revisions are listed as follows point-by-point.

Specific comment 1. *Generally, the paper was difficult to read due to multiple referral to the supplementary information document, thereby affecting the overall flow, but I understand this pertains mainly to restricted number of words.*

Our responses: Because of the restricted paper length, we put some data and results in the supplementary files and only the main results are shown in text. In the revised manuscript, reference to Supplementary information (SI) has been reduced to keep clarity. To improve the readability and flow of this paper, we deleted some tables in the SI (Supplementary Tables 1-9 in the previous version) if the related data have been presented or partially presented in figures and some tables related to the calculation process (Supplementary Tables 22-26 in the previous version). Accordingly, all other tables are renumbered, and the related expressions have been revised in both text and SI.

Specific comment 2. *In line 56, the expression “CCS is the only option that would ultimately achieve a deep reduction in CO₂ emissions from this sector” is a bit of an overstatement. Different technology options offers different emission reduction potentials depending on the scenarios under consideration. The use of scrap-based EAFs with low carbon electricity has been touted as a viable option. High project costs, limited geological storage and public scrutiny has stalled development of CCS. These issues must be taken into consideration when assessing CCS whether as a standalone entity or as part of a wider systems. For CCS to be deployed at a commercial level, numerous issues including cost of implementation, monitoring and validation, regulations and legal aspects as well as public acceptance must resolved. Authors are encouraged to note these factors in their manuscript.*

Our responses: Regarding the emission reduction toward sectoral 2 °C target in the iron and steel industry, many options have been proposed. These options could be divided into internal technological developments like scrap/EAF and DRI and external technological solutions coupled to steel like CCS. Since the current study is discussed in a long range to 2050, all these options can contribute to the 2 °C target. CCS is in the basket for emission reduction but not the only option due to its own drawbacks, as mentioned by the Reviewer. To clarify the foregoing points, the expressions that “CCS is the only option that would ultimately achieve a deep reduction in CO₂ emissions from this sector” have been revised as “CCS remains a critical option that could ultimately achieve a deep reduction in CO₂ emissions from this sector^{5,10}, although its costs and externalities remain to be assessed in practical industrial cases for coupling CCS facilities with steel production.” in lines 54-56.

Specific comment 3. *Steel slags also finds a wide range of applications including agricultural fertilizers, road construction, soil improvements, etc. Authors are encouraged to briefly comment on this aspect and make a case as to why CCS is the preferred approach in comparison to other applications.*

Our responses: We agree that steel slag has found a wide range of applications, including agricultural fertilizers, soil improvements, road constructions, etc., after heat and valuable CaO has been separated and re-used^{1,2}. In this study, we do not exclude these utilizations. These final utilizations could also be considered alongside with CCS. To clarify this point, the

sentences related to the further utilization of steel slag “Recently, other applications of SS such as agricultural fertilizers²⁹ and soil improvement agents³⁰ have also been developed.” have been added in **lines 232-233**. Accordingly, new **references 29** and **30** have been added into the **reference list** in **lines 554-557**.

1. Das, S., Kim, G.W., Hwang, H.Y., Verma, P.P. & Kim, P.J. Cropping with slag to address soil, environment, and food security. Front. microbiol. 10, 1320 (2019). (Numbered 29 in Revised Manuscript)

2. Ito, T., Nasu, K., Saito, M. & Kitamura, S. Productivity improvement of saline paddy soils caused by seawater inflow with steelmaking slag fertilizer. CAMP-ISIJ 27, 322 (2014). (Numbered 30 in Revised Manuscript)

Specific comment 4. *Lines 281 to 283, the authors stated, “Therefore, it is assumed that the annual BFS/Pig iron ratio and SS/Crude steel ratio will decrease at stable rates of 0.002/year and 0.001/year from 2020 to 2050, respectively”. On what basis were these assumptions made?*

Our responses: The discharge ratios of BFS/Pig iron and SS/Crude steel are assumed based on data collected in the past years. There are very limited research papers accurately reporting the discharge of BFS and SS or predicting the future BFS and SS productions especially in the mid- to long-term.

For example, regarding the BFS/Pig iron ratio in the past years, we can use the outputs of pig iron collected by the World Steel Association¹ and discharges of BFS summarized from the Global Slag Review^{2,3}. In 2018, it was reported 333 Mt BFS was discharged² with the pig iron output of 1253 Mt¹; it can be calculated that the BFS/Pig iron in 2018 was around 0.266. In 2017, it was reported 360 Mt BFS was discharged² with a pig iron output of 1186 Mt¹, which gives a BFS/Pig iron ratio of 0.304. Based on these estimations, we can see the great decrease of BFS/Pig iron ratio. The discharges of BFS and SS were also collected from the U.S. Geological Survey⁴. However, a large range is provided; for example, 320~384 Mt BFS and 190-280 Mt SS were discharged in 2019; accordingly, the ranges of BFS/Pig iron and SS/Crude steel were 0.25~0.30 and 0.10~0.15, respectively. In this study, a decreasing and stable BFS/Pig iron ratio was used to simplify the analysis considering possible technological

progress to reduce this ratio in the future (similarly to SS/Crude steel ratio).

To clarify the foregoing analysis, **firstly**, the sentences “*In the current manufacturing process of the iron and steel sector, producing one tonne of crude steel generates considerable waste streams, including 250-350 kilograms of blast furnace slag (BFS) at temperatures of 1500-1600 °C and 100-200 kilograms of steel slag (SS) at temperatures of 1550-1650 °C (ref. ¹²).*” have been revised as “**In the current manufacturing process of iron and steel, producing one tonne of crude steel generates waste streams in the range of 250-300 kilograms of blast furnace slag (BFS) at temperatures of 1500-1600 °C and 100-150 kilograms of steel slag (SS) at temperatures of 1550-1650 °C¹¹⁻¹⁴.**” in **lines 61-64**. **Secondly**, the sentences “*At the current global level in 2020, these two parameters are assumed to be approximately 0.260 and 0.130, respectively¹²⁻¹⁵. Due to technological advancements, the discharge of BFS and SS will decrease continuously¹³⁻¹⁵. Therefore, it is assumed that the annual BFS/Pig iron ratio and SS/Crude steel ratio will decrease at stable rates of 0.002/year and 0.001/year from 2020 to 2050, respectively.*” have been revised as “**At the current global level in 2020, these two parameters are approximately 0.26 and 0.13, respectively¹¹⁻¹⁴. Due to technological progress, the discharge of BFS and SS could decrease continuously^{12,13}. Therefore, it is assumed that the BFS/Pig iron and SS/Crude steel ratios will decrease at rates of 0.002/year and 0.001/year from 2020 to 2050, respectively¹¹⁻¹⁴.**” in **lines 272-276**. Moreover, the references have been updated correspondingly.

1. World Steel Association, *Statistical reports*,

<https://www.worldsteel.org/steel-by-topic/statistics/steel-statistical-yearbook.html> (Numbered 14 in Revised Manuscript)

2. Global Slag, *Global Slag 2018 Review*:

<https://www.globalslag.com/conferences/global-slag/review/global-slag-review-2018>
(Numbered 12 in Revised Manuscript)

3. Global Slag, *Global Slag 2017 Review*:

<https://www.globalslag.com/conferences/global-slag/review/global-slag-review-2017>
(Numbered 13 in Revised Manuscript)

4. U.S. Geological Survey,
<https://www.usgs.gov/centers/nmic/iron-and-steel-slag-statistics-and-information?qt-science>

Specific comment 5. *In terms of the cost benefit analysis (CBA) presented, authors listed all the parameters (e.g. capital, labour, energy maintenance etc) taken into consideration (expanded upon in the SI), yet there is no description or any mention of the actual type of CBA conducted and the equations linking all the parameters together. As such, it is difficult to ascertain how the CB figures were calculated. Were the calculations based on life cycle costing? Or on principles of marginal abatement cost curves (MACC)? Were the net present values (NPV) of the potential cost savings taken into consideration in the CBA? Overall, it is not clear at all the actual CBA framework adopted by the authors. Parameters for the CBA were listed and described but how they combine based on a defined calculation procedure is not stated. The calculation steps might be obvious to the authors in the supplementary figures, but may not be clear to the wider readership of Natur Comms. Essentially, an equation linking all the cost parameters together must be provided using an appropriate mathematical equation. For a paper that expresses cost gains as part of the key findings, the CBA presented is not rigorous enough.*

Our responses: We thank the Reviewer for this valuable comment. The net revenue per tonne of crude steel is calculated based on those per tonne of BFS and SS. To address the Reviewer's comment, we added a mathematical equation to clearly show the relationship. The use of this equation is also good for the readers to understand the economics and the economic sensitivity, which is very important. The equation is shown as follows:

$$NR = \alpha[(B_{BFS1} + B_{BFS2} + \dots + B_{BFSn}) - (C_{BFS1} + C_{BFS2} + \dots + C_{BFSn})] \\ + \beta\gamma[(B_{SS1} + B_{SS2} + \dots + B_{SSn}) - (C_{SS1} + C_{SS2} + \dots + C_{SSn})]$$

where α , β and γ are the BFS to pig iron, the SS to crude steel and the pig iron to crude steel ratios, respectively; B_{BFS} and C_{BFS} are the different benefits and costs per tonne of BFS during the treatments (same for the SS subscript).

To clarify the foregoing points, **firstly**, the equation and the related expressions have been added in **lines 468-475**. **Secondly**, the related sentences “*To further estimate the output of pig iron, the weight ratio of pig iron to crude steel (Pig iron/Crude steel) should be determined.*”

have been revised as “To further estimate the output of pig iron, the weight ratio of pig iron to crude steel (Pig iron/Crude steel, named as α) should be determined.” in lines 260-261.

Thirdly, the related sentences “After the global outputs of pig iron and crude steel are obtained, two other parameters are required to obtain the global production of BFS and SS, namely, the weight ratio of BFS to pig iron (BFS/Pig iron) and that of SS to crude steel (SS/Crude steel).” have been revised as “After the global outputs of pig iron and crude steel are obtained, two other parameters are required to obtain the global productions of BFS and SS, namely the weight ratio of BFS to pig iron (BFS/Pig iron, named as β) and that of SS to crude steel (SS/Crude steel, named as γ).” in lines 270-272.

Specific comment 6. *The style and language in which the Methods section was written should be revised. For instance, the authors stated: “To estimate the CO₂ emission reductions in different pathways in the timeframe of 2020-2050, the global outputs of crude steel should first be estimated”. This and other statements should be revised.*

Our responses: The style and language in the Methods section have been revised. The sentences pointed by the Reviewer have been revised as “To estimate the potential of CO₂ emission reductions in different pathways in 2020-2050, the global outputs of crude steel should be first predicted (Supplementary Table 3).” in lines 257-258. Additionally, all the statements have been checked and revised to improve the readability and we have also asked the language editing company to help us to improve the language.

Specific comment 7. *The calculation steps for the estimation of CO₂ emission reductions based on energy recovery from BFS and SS is basic. Authors are encouraged to adopt a more robust energy balance approach in this regard.*

Our responses: Regarding the estimation of CO₂ emission reductions based on energy recovery from BFS and SS, it is mainly based on the integration of heat capacity and temperature, following mass and energy balances. In this study, we use two methods to calculate the heat capacities, namely linear addition of various oxides and direct calculation. Both are tested in this study and the results are similar. The calculation process is detailed in Supplementary Note 6. To clarify the foregoing point, the sentences “There are two methods

to obtain the heat capacities of BFS and SS (details in Supplementary Note 6).” have been revised as “There are two methods to obtain the heat capacities of BFS and SS (Supplementary Note 6) based on the basic principles of mass and energy balances.” in lines 287-289.

Specific comment 8. *I am not sure about the requirements for supplementary document by Nature Comms, however, in the current submission, the supplementary document is 95 pages long. The first half of the paper is laced with linkages to the supplementary document, affecting flow and readability. The supplementary document should be as a supplement (as the name suggests) and not as the primary repository for the analysis and figures. If this is in line with the journal’s policy, please ignore this comment.*

Our responses: As pointed by the Review in Specific comment 1, due to the limitation of paper length, we put some data and results in the supplementary files and only the main results are shown in text. To improve the readability and flow of this paper, we deleted some tables in SI if the related data have been presented or partially presented in figures (detailed in the responses to Specific comment 1).

Specific comment 9. *In some portions of the paper, there are minor typos, which should be revised (e.g. raise instead of rise (line 4, pg. 2). Please correct all typos.*

Our responses: We sincerely thank the Reviewer for the careful work. We have checked the whole manuscript and revised the typos in the text, figures, and tables (like labor to labour, slags to slag, etc.), as detailed in the Revised Manuscript with Track Changes.

Peer Review File, further round reviewer comments –

Reviewer #1 (Remarks to the Author):

I suggest that the method you propose be called recovery of exergy or rather avoiding exergy destruction, at energy and chemical composition levels.

The targets for the steel sector today are NET-ZERO emissions. See for example the Green Deal of the EC.

I disagree with your comments p.8 regarding the interest of the European steel sector in handling slag, actually in slagmaking, as someone I know coined the expression. The work done in the EU was carried out 20 years ago and clearly your bibliographical review does not go that deep in time. The Asian papers you quote are actually Europe-blind and did not take them on board, hence a duplication of efforts at world level. This is not related to the fact that the production of steel in Asia (actually in China) has overcome by almost an order of magnitude that of Europe.

At the end of the day, you have watered down your initial claims, adding caveats regarding the actual ability to implement your pathways. I feel you still rely on solutions which are not yet mature enough and which I do not believe will probably never mature in time.

There are more promising solutions for going to net-zero steelmaking. You should at the very least pay lip service to them.

Reviewer #2 (Remarks to the Author):

The authors have revised the paper according to the suggestions provided by reviewers, by enriching it with some additional contents which allow overcoming some weaknesses highlighted in previous review reports.

Reviewer #3 (Remarks to the Author):

From my perspective, the authors have satisfactorily addressed my comments. I however have reservation on the potential cost savings reported by the author: "the technological schemes applied to engineer this high-potential pathway could generate a revenue of US\$35 and US\$40 billion globally in 2035 and 2050, respectively".

For a cost benefit analysis that does not apply net present value despite projecting cost benefits up to 2035 and 2050, the benefits reported are laced with uncertainty. There is no doubt regarding the inherent financial benefits of recovering energy and reusing materials from waste streams and high-temperature slag, however, the figures reported are ambitious. I therefore advise the authors to revise this statement using approximate figures and +-uncertainty. Currently, the chances that CCS will become feasible at a global level is low but with the right policy support and financial incentives (e.g., as in the case reported by the authors), its adoption and implementation will become viable. So, this is a welcome addition.

Other than the cost figures reported with certainty, the analysis presented is rigorous and addresses an important research area.

Responses to the comments from Reviewer #1

Comment 1: *I suggest that the method you propose be called recovery of exergy or rather avoiding exergy destruction, at energy and chemical composition levels.*

Our responses: We sincerely thank the Reviewer for all the valuable comments, suggestions, and discussions. As the Reviewer suggested, the treatment of waste streams could be understood as an exergy recovery process, especially for the heat recovery from high temperature slag¹⁻³. To clarify this point, the sentence “*The high-temperature slag contains energy and resources, offering a large internal recovery potential and the revenues that could be re-invested into CCS to lower its cost*¹⁵⁻¹⁷.” has been revised as “**The high-temperature slag contains high-degree exergy at the levels of thermal energy and material resources, offering a large internal recovery potential to generate revenues that could be re-invested into CCS to lower its cost**¹⁶⁻¹⁸.”, in **lines 70-72**. In addition, to keep the consistence of the expressions and good readability, we still use heat recovery and material recycling in most cases.

References:

1. Bisio, G. *Energy recovery from molten slag and exploitation of the recovered energy. Energy* 22, 501-509 (1997). (Numbered 18 in Revised Manuscript)
2. Maruoka, N., Purwanto, H. & Akiyama, T. *Exergy analysis of methane steam reformer utilizing steelmaking waste heat. ISIJ Int.* 50: 1311-1318 (2010).
3. Duan W, et al. *The technological calculation for synergistic system of BF slag waste heat recovery and carbon resources reduction. Energy Convers. Manag.* 87:185-90 (2014).

Comment 2: *The targets for the steel sector today are NET-ZERO emissions. See for example the Green Deal of the EC.*

There are more promising solutions for going to net-zero steelmaking. You should at the very least pay lip service to them.

Our responses: More countries and regions have set the net-zero emission targets and the roadmaps including Europe (2050), Australia (2050), China (2060), etc. These carbon-neutral targets come with requirements for carbon emission reduction for the iron and steel industry. Regarding the *Green Deal of the EC* pointed by the Reviewer, the plan is to reduce carbon emission by 55% by 2030 and by 80-95% by 2050 (compared to 1990) towards carbon

neutrality for the iron and steel industry^{1,2}, which is more ambitious than the 2 °C emission profile in refs³⁻⁵. The achievement of this target greatly relies on deployment of breakthrough technologies including Carbon Direct Avoidance (CDA: hydrogen- and electricity-based metallurgy), and Smart Carbon Usage (SCU: Process integration and Carbon Valorisation, CV, Carbon Capture and Usage, CCU), in addition to the technologies discussed in our study such as treatment of waste streams, keeping use of scrap-based EAF steelmaking, top gas pressure recovery turbines (TRTs), and Carbon Capture and Storage (CCS).

To reflect the up-to-date targets, we have added more information on the Green Deal of the EC and the zero-emission steel target. **Firstly**, in the **Introduction** section, the sentence “Furthermore, towards the net-zero emission steel in the long term like the EU Green Deal, more breakthrough technologies should be deployed such as hydrogen- and green electricity-based metallurgy and smart carbon usage (Process integration and Carbon Valorisation, Carbon Capture and Usage-CCU, etc)¹¹.” has been added in the manuscript, in **lines 62-65**. **Secondly**, in the **Discussion** section, the sentence “Moreover, the pathways and technology schemes discussed in this study could be an important wedge for approaching net-zero emissions for the steel industry as proposed by the EU Green Deal¹¹, along with other transformational changes such as hydrogen- and green electricity-based metallurgy and smart carbon usage¹¹.” has been added in the manuscript, in **lines 255-258**. Accordingly, **reference 11** has been added in the reference list, in **lines 534-537** and all other references have been renumbered.

References:

1. *The European Steel Association, A Green Deal on Steel, https://www.eurofer.eu/assets/publications/brochures-booklets-and-factsheets/we-are-ready-are-you-making-a-success-of-the-eu-green-deal/20210122-EUROFER_Making-a-Success-of-the-EU-Green-Deal.pdf (Numbered 11 in Revised Manuscript)*
2. *The European Steel Association, LOW CARBON ROADMAP PATHWAYS TO A CO₂-NEUTRAL EUROPEAN STEEL INDUSTRY, <https://www.eurofer.eu/assets/Uploads/EUROFER-Low-Carbon-Roadmap-Pathways-to-a-CO2-neutral-European-Steel-Industry.pdf>*
3. *International Energy Agency. Energy Technology Perspectives 2016: Towards Sustainable*

Urban Energy Systems. <https://www.iea.org/topics/energy-technology-perspectives>.

(Numbered 4 in Revised Manuscript)

4. Tian, S., Jiang, J., Zhang, Z. & Manovic, V. Inherent potential of steelmaking to contribute to decarbonisation targets via industrial carbon capture and storage. Nat. Commun. 9, 4422 (2018). (Numbered 5 in Revised Manuscript)

5. Birat, JP., ArcelorMittal Global R & Maizières-lès-Metz. D., UNIDO. Steel Sectoral Report: Contribution to the UNIDO Roadmap on CCS –Fifth Draft. (Numbered 6 in Revised Manuscript)

Comment 3: *I disagree with your comments p.8 regarding the interest of the European steel sector in handling slag, actually in slagmaking, as someone I know coined the expression. The work done in the EU was carried out 20 years ago and clearly your bibliographical review does not go that deep in time. The Asian papers you quote are actually Europe-blind and did not take them on board, hence a duplication of efforts at world level. This is not related to the fact that the production of steel in Asia (actually in China) has overcome by almost an order of magnitude that of Europe.*

Our responses: The metallurgical slag could be reused for slagmaking when the final chemical compositions are well controlled by phase separation and impurity removals to meet the basic requirement of ironmaking and steelmaking such as dephosphorization, desulfurization, and deoxidization¹⁻⁶. During the ironmaking and steelmaking, fluxes with designed compositions, e.g., CaO, CaF₂, MgO, B₂O₃, etc., will react with the liquid alloy to remove impurities (S, P₂O₅, etc.) and meanwhile, they will react with the newly formed oxides such as Fe₂O₃, SiO₂, Al₂O₃, etc., to form final slag with target compositions and thermal physical properties^{7,8}. This is slagmaking process in the steel industry. Therefore, slagmaking reactions greatly determine the final compositions of hot slag and therefore the treatment strategies. Extensive work has been done regarding the heat recovery and resource recycling of the hot slag¹⁻⁶.

Europe have made important contributions for the slag treatment including those for slagmaking¹⁻⁶. For example, for the heat recovery from BFS, Pickering et al.¹ developed a dry granulation method to granulate the hot BFS into small droplets to increase the heat exchange

with working medium. For the resource recycling of slag²⁻⁶, the recirculation of BOF slag as fluxes for slagmaking in BF or BOF has been tried and realized commercialization in Europe. And J. Geiseler^{4,5} summarized that 17% BOF slag and 4% EAF slag are recycled for further metallurgical use in Europe, with the remaining slag used for road pavements, waterway construction, fertilizer, etc. To clarify the European contributions on slag treatment and slagmaking, we have made several revisions.

Firstly, in the **Introduction** section, the sentence “*From the resource point of view, both BFS and SS contain more than 40 wt. % CaO fluxed from limestone calcination^{18,19}, the recycling of which constitutes a significant Ca-source, e.g. for cement industry that could to reduce its CO₂ emissions²⁰⁻²².*” has been revised as “**From the resource point of view, both BFS and SS contain more than 40 wt. % CaO fluxed from limestone calcination^{19,20}, the recycling of which constitutes a significant Ca-source, e.g., for slagmaking in metallurgy²¹⁻²³ and for cement production to reduce their CO₂ emissions²⁴⁻²⁶.**”, **in lines 75-78**. **Secondly**, the sentence “*... high economic-value reuse of CaO to replace limestone calcination in the cement or steel sector after necessary iron and phosphorus separations (SS-Crystalline/Dry-R)¹⁶⁻¹⁸.*” has been revised as “**... high economic-value reuse of CaO to replace limestone calcination in the cement or steel sector like for slagmaking after necessary iron and phosphorus separations (SS-Crystalline/Dry-R)^{16-18,21-23}.**”, **in lines 117-119**. **Thirdly**, the sentence “**For the reuse of slag in the iron and steel industry like for slagmaking, the final chemical compositions should be well controlled by necessary phase separations and impurity removals²¹⁻²³.**” has been added in the **Discussion** section **in lines 243-245**. **Fourthly**, the **references 21-23** on the treatment of slag in Europe especially those for slagmaking have been added in the reference list **in lines 558-562** and all other references have been renumbered.

References:

1. Pickering, S.J., Hay, N., Roylance, T.F. & Thomas, G.H. *New process for dry granulation and heat recovery from molten blast-furnace slag. Ironmak. Steelmak. 12(1), 14-21 (1985).* (Numbered 21 in Revised Manuscript)
2. Ökvist, L.S. *High temperature properties of BOF slag and its behaviour in the blast furnace. Steel Res. Int. 75(12), 792-799 (2004).* (Numbered 22 in Revised Manuscript)
3. Björkman, B., Eriksson, J., Nedar, L. & Samuelsson, C. *Waste reduction through process*

optimization and development. JOM 48(3), 45-49 (1996).

4. Geiseler, J. *Use of steelworks slag in Europe. Waste Manage. 16(1-3), 59-63 (1996).*

(Numbered 23 in Revised Manuscript)

5. Motz, H. & Geiseler, J. *Products of steel slags an opportunity to save natural resources.*

Waste Manage. 21(3), 285-293 (2001).

6. Bisio, G. *Energy recovery from molten slag and exploitation of the recovered energy.*

Energy 22, 501-509 (1997). (Numbered 18 in Revised Manuscript)

7. Iyengar, R.K. & Petrilli, F.C. *Slagmaking reactions in the BOF process. JOM 25(7), 21-26*

(1973).

8. Smirnov, L.A. *Characteristics of slag-making and metal-refining processes in oxygen-blown*

converters for irons of special compositions. Can. Metall. Q 22(3), 305-312 (1983).

Comment 4: *At the end of the day, you have watered down your initial claims, adding caveats regarding the actually ability to implement your pathways. I feel you still rely on solutions which are not yet mature enough and which I do not believe will probably never mature in time.*

Our responses: As the Reviewer acknowledged, we added uncertainty ranges of pathway potential and economics based on a sensitivity analysis to facilitate understanding and interpretation of the results. The pathways and technology schemes in this study are constructed strictly based on the fundamental properties of BFS and SS and current technological levels while considering future technological progress. For the technologies involved in those pathways and schemes, their development states are quite different regarding BFS and SS. Some of them have realized good commercial utilization while others are tested in pilot-scale or lab-scale. We think it is possible for the technology solutions discussed in this study to be mature in the long-term decarbonization process in the iron and steel industry.

For BFS, three main approaches are developed to cool hot slag, natural cooling, water quenching, and dry granulation. The water-quenching process combined with cement production has been realized good commercial utilizations¹⁻³. However, thermal heat in hot BFS is normally wasted. Therefore, obtaining the glassy state with effective heat recovery is

the most promising proposed technology, i.e., the integration of heat recovery and resource recycling. Along with time going, lab-scale technologies are expected to become mature and industrialized. For example, using the thermal heat in BFS to produce slag wool has already been commercialized⁴ and dry granulation of BFS for heat recovery and resource recycling is tried at pilot scale in Australia⁵. The technological solutions for BFS treatment are summarized in **lines 104-108**, detailed as “Currently, three main approaches have been proposed to cool hot BFS with heat recovery and resource recycling, namely natural cooling, water quenching, and dry granulation, with quite different practical cooling rates and states of the cooled BFS. Accordingly, three strategies are proposed for BFS treatment (Supplementary Note 1): BFS-Glassy/Water, BFS-Glassy/Dry, and BFS-Crystalline/Dry.” In addition, the development states, challenges, and future progresses of those solutions for BFS treatments are further discussed in **Supplementary Note 1**.

As for SS including BOF and EAF slag, it is not a perfect match for cement making with greater challenges because of its chemical compositions like high basicity (CaO/SiO₂) and “FeO” concentration. We believe that future treatment methods could be multiple while the main targets are to reuse the valuable compositions in the SS like CaO, “FeO” and P₂O₅.^{6,7} The heat recovery from SS is another big challenge because of its high viscosity and crystallization behaviours. That is why in our study, cooled SS is assumed to be transformed in the crystalline state in both Pathways 5 and 6. Recently, there are also some exciting progresses; for example, Tobo et al.⁸, developed continuous solidification process for sensible heat recovery from SS and Zhang et al.⁹, conducted industrial-scale tests on the heat recovery from SS. The slag after necessary phase separations and impurity removals could also be reused for slagmaking, as discussed in the response to **Comment 3**. The solutions for SS treatment are summarized in **lines 119-121**, detailed as “Summarizing these characteristics, three strategies are proposed for SS treatment (Supplementary Note 2): SS-Crystalline/Dry, SS-Crystalline/Dry-R, and SS-Glassy/Water.” In addition, the development states, challenges, and future progresses of those solutions are further discussed in **Supplementary Note 2**.

References:

1. Barati, M., Esfahani, S. & Utigard, T. A. *Energy recovery from high temperature slags. Energy* 36(9), 5440–5449 (2011). (Numbered 15 in Revised Manuscript)

2. Zhang, H. et al. A review of waste heat recovery technologies towards molten slag in steel industry. *Appl. Energy* 112, 956–966 (2013). (Numbered 16 in Revised Manuscript)
3. Bisio, G. Energy recovery from molten slag and exploitation of the recovered energy. *Energy* 22, 501-509 (1997). (Numbered 17 in Revised Manuscript)
4. Zhao, D., Zhang, Z., Tang, X., Liu, L. & Wang, X. Preparation of slag wool by integrated waste-heat recovery and resource recycling of molten blast furnace slags: from fundamental to industrial application. *Energies* 7, 3121-3135 (2014).
5. Cooksey, M., Guiraud, A., Kuan, B. & Pan, Y. Design and operation of dry slag granulation pilot plant. *J. Sustain. Metall.* 5, 181-194 (2019).
6. Engström, F., Adolfsson, D., Yang, Q., Samuelsson, C. & Björkman, B. Crystallization behaviour of some steelmaking slags. *Steel Res. Int.* 81, 362-371 (2010). (Numbered 24 in Revised Manuscript)
7. Yokoyama, K. et al. Separation and recovery of phosphorus from steelmaking slags with the aid of a strong magnetic field. *ISIJ Int.* 47, 1541-1548 (2007). (Numbered 28 in Revised Manuscript)
8. Tobo, H. et al. Development of continuous steelmaking slag solidification process suitable for sensible heat recovery. *ISIJ Int.* 55, 894–903 (2015).
9. Zhang, T. et al. In-Suit Industrial Tests of the Highly Efficient Recovery of Waste Heat and Reutilization of the Hot Steel Slag. *ACS Sustain. Chem. Eng.*, 9, 3955-3962 (2021).

Responses to the comments from Reviewer #2

General comments:

The authors have revised the paper according to the suggestions provided by reviewers, by enriching it with some additional contents which allow overcoming some weaknesses highlighted in previous review reports.

Our responses: We sincerely thank the Reviewer for the positive comments and all the constructive suggestions.

Responses to the comments from Reviewer #3

Comment 1. *From my perspective, the authors have satisfactorily addressed my comments. I however have reservation on the potential cost savings reported by the author: “the technological schemes applied to engineer this high-potential pathway could generate a revenue of US\$35 and US\$40 billion globally in 2035 and 2050, respectively”.*

For a cost benefit analysis that does not apply net present value despite projecting cost benefits up to 2035 and 2050, the benefits reported are laced with uncertainty. There is no doubt regarding the inherent financial benefits of recovering energy and reusing materials from waste streams and high-temperature slag, however, the figures reported are ambitious. I therefore advise the authors to revise this statement using approximate figures and +-uncertainty.

Other than the cost figures reported with certainty, the analysis presented is rigorous and addresses an important research area.

Our responses: We sincerely thank the Reviewer for the valuable comments and suggestions. To address the comments by the Reviewer, we **firstly** conduct a systemic analysis on the economic sensitivity of the five technology schemes in **lines 195-207**, detailed as “**Economic uncertainty originates from the varying product price, process and carbon price, estimated by sensitivity studies. Figure 2d (Supplementary Figs. 15-18) shows that for all schemes, the net revenue is mostly sensitive to the slag price because cooled slag (glassy BFS in particular) accounts for the dominant valuable product. If the slag price increases by 50%, the revenue increases from US\$15 to US\$22 million, while if the slag price decreases by 50%, the total revenue decreases to US\$8 million. As a result, the formation of BFS in the glassy state appears to be a key target for BFS treatment for all the proposed schemes. Comparatively, the effects of labour, maintenance and energy costs are smaller than the revenues from better re-use of slag products in our analysis. We acknowledge uncertainties in estimating labour and maintenance for slag recovery technologies, as these solutions are not implemented at scale**

today. In Scheme 1, the capital cost, steam price and assumed CO₂ price play comparable roles. In Schemes 2-5, the syngas price also remarkably affects the plant economics as syngas is another valuable product, while the effect of the gasification fuel price is limited.” The sensitivity analysis is conducted at a steel plant level, which also works for the global level because they show the same sensitivity variables and ratios. **Secondly**, the sentence “**That could be further calculated at the global level based on the economic sensitivity at the level of a steel plant, since they show the same sensitivity variables and ratios.**” has been thus added in the manuscript, in **lines 207-209**. **Thirdly**, following the Reviewer’s suggestion, we have revised the net benefit with uncertainty ranges in the whole paper, including **lines 32, 88, 220 and 264**. **Fourthly**, the sentence “**Here we have shown ambitious figures, with right policy support and assuming no barrier to scalability, technical progress allowing industrialization, and effectiveness of CCS attached to steel facilities.**” has been added in the revised manuscript, in **lines 228-231**.

Comment 2. *Currently, the chances that CCS will become feasible at a global level is low but with the right policy support and financial incentives (e.g., as in the case reported by the authors), its adoption and implementation will become viable. So, this is a welcome addition.*

Our responses: As the Reviewer pointed, right policy support and financial incentives are very important for the global feasibility of CCS and its deep integration with treatment of waste streams and other technology solutions in the iron and steel industry. We have added a sentence to clarify this point in the revised manuscript, as detailed in the response to **Comment 1**.

Peer Review File, final reviewer comments–

Reviewer #2 (Remarks to the Author):

Concerning the new parts added by authors after the reviewers comments, some specific comments can be provided:

- In lines 62-65 and lines 255-258 authors mentioned the European Green Deal and the steel sector' commitments to achieve its targets. However, these aspects are only mentioned, including breakthrough technologies, but more examples and references are missing in particular in the Introduction section.

- In order to support new added contents on the treatment of slag in Europe, new references were added, such as 21, 22, 23. However, the studies cited are a bit old, while more recent works should be included and discussed, as these topics have been recently studied.

Reviewer #3 (Remarks to the Author):

The authors have further improved the manuscript following my suggested ideas. From my perspective, the manuscript has met all criteria to recommend it for publication

Responses to the comments from Reviewer #2

General comment: *Concerning the new parts added by authors after the reviewers comments, some specific comments can be provided.*

Our responses: We sincerely thank the Reviewer for the further comments and all the constructive suggestions for previous versions. Accordingly, we have updated the references as well as the related expressions.

Comment 1: *In lines 62-65 and lines 255-258 authors mentioned the European Green Deal and the steel sector' commitments to achieve its targets. However, these aspects are only mentioned, including breakthrough technologies, but more examples and references are missing in particular in the Introduction section.*

Our responses: As the Reviewer said, in lines 62-65 and line 255-258, we introduced the European Green Deal as a good example for the future zero-emission target in the iron and steel industry. To achieve this ambitious target, a series of breakthrough technologies are needed including hydrogen- and green electricity-based metallurgy and smart carbon usage (Process integration and Carbon Valorisation, Carbon Capture and Usage-CCU, etc)¹⁻³, as well as the recovery/recycling of the waste streams in metallurgy discussed in this study. According to the Reviewer's comment, we have provided specific references for those technologies and added references 12 and 13 have been added. Correspondingly, all other references have been renumbered.

In addition, the sentence "*Furthermore, towards the net-zero emission steel in the long term like the EU Green Deal, more breakthrough technologies should be deployed such as hydrogen- and green electricity-based metallurgy and smart carbon usage (Process integration and Carbon Valorisation, Carbon Capture and Usage-CCU, etc)*¹¹." have been revised as "**Furthermore, the net-zero emission steel target in the long term like the EU Green Deal¹¹ necessitates more breakthrough technologies such as hydrogen- and green electricity-based metallurgy and smart carbon usage (Process integration and Carbon Valorisation, Carbon Capture and Usage-CCU, etc)¹¹⁻¹³.**" in **lines 62-65**.

References:

1. *The European Steel Association, A Green Deal on Steel:*

https://www.eurofer.eu/assets/publications/brochures-booklets-and-factsheets/we-are-ready-are-you-making-a-success-of-the-eu-green-deal/20210122-EUROFER_Making-a-Success-of-the-EU-Green-Deal.pdf (Numbered 11 in Revised Manuscript)

2. Tang, J., et al. Development and progress on hydrogen metallurgy. *Int. J. Miner. Metall. Mater.* 27, 713-723 (2020). (Numbered 12 in Revised Manuscript)

3. Vogl, V., Åhman, M. & Nilsson, L. J. Assessment of hydrogen direct reduction for fossil-free steelmaking. *J Clean Prod.* 203, 736-745 (2018). (Numbered 13 in Revised Manuscript)

Comment 2: In order to support new added contents on the treatment of slag in Europe, new references were added, such as 21, 22, 23. However, the studies cited are a bit old, while more recent works should be included and discussed, as these topics have been recently studied.

Our responses: We sincerely thanks the Reviewer for the valuable suggestions. Accordingly, we have added recent studies (Branca *et al.*, 2014; Skaf *et al.*, 2017; Hadj *et al.*, 2021) on the treatment of slag including BOF¹, EAF² and BFS³ in Europe, in addition to the previous progresses. Correspondingly, all other references have been renumbered.

References:

1. Branca, T.A., et al. Investigation of (BOF) Converter slag use for agriculture in europe. *Metall. Res. Technol.* 111, 155-167 (2014). (Numbered 26 in Revised Manuscript)

2. Skaf, M., Manso, J. M., Aragón, Á., Fuente-Alonso, J. A. & Ortega-López, V. EAF slag in asphalt mixes: A brief review of its possible re-use. *Resour. Conserv. Recycl.* 120, 176-185 (2017). (Numbered 27 in Revised Manuscript)

3. Hadj Sadok, R., et al. Mechanical properties and microstructure of low carbon binders manufactured from calcined canal sediments and ground granulated blast furnace slag (GGBS). *Sustainability* 13, 9057 (2021). (Numbered 28 in Revised Manuscript)

Responses to the comments from Reviewer #3

General comment: *The authors have further improved the manuscript following my suggested ideas. From my perspective, the manuscript has met all criteria to recommend it for publication.*

Our responses: We sincerely thank the Reviewer for the positive comments and all the constructive suggestions.